# Phase response analyses support a relaxation oscillator model of locomotor rhythm generation in *Caenorhabditis elegans*

**Hongfei Ji[1†], Anthony D Fouad[1†], Shelly Teng[1], Alice Liu[1], Pilar Alvarez-Illera[1], Bowen Yao[1], Zihao Li[1], Christopher Fang-Yen[1,2]***

[1]Department of Bioengineering, School of Engineering and Applied Science, University of Pennsylvania, Philadelphia, United States; [2]Department of Neuroscience, Perelman School of Medicine, University of Pennsylvania, Philadelphia, United States

**Abstract** Neural circuits coordinate with muscles and sensory feedback to generate motor behaviors appropriate to an animal's environment. In *C. elegans*, the mechanisms by which the motor circuit generates undulations and modulates them based on the environment are largely unclear. We quantitatively analyzed *C. elegans* locomotion during free movement and during transient optogenetic muscle inhibition. Undulatory movements were highly asymmetrical with respect to the duration of bending and unbending during each cycle. Phase response curves induced by brief optogenetic inhibition of head muscles showed gradual increases and rapid decreases as a function of phase at which the perturbation was applied. A relaxation oscillator model based on proprioceptive thresholds that switch the active muscle moment was developed and is shown to quantitatively agree with data from free movement, phase responses, and previous results for gait adaptation to mechanical loadings. Our results suggest a neuromuscular mechanism underlying *C. elegans* motor pattern generation within a compact circuit.

**\*For correspondence:**
fangyen@seas.upenn.edu

[†]These authors contributed equally to this work

**Competing interests:** The authors declare that no competing interests exist.

## Introduction

Animal display locomotor behaviors such as crawling, walking, swimming, or flying via rhythmic patterns of muscle contractions and relaxations. In many animals, motor rhythms originate from networks of central pattern generators (CPGs), neuronal circuits capable of generating rhythmic outputs without rhythmic input (*Cohen and Wallen, 1980*; *Grillner, 2003*; *Kiehn, 2011*; *Kristan and Calabrese, 1976*; *Marder and Calabrese, 1996*; *Pearce and Friesen, 1984*; *Yu et al., 1999*). CPGs typically generate rhythms through reciprocal inhibitory synaptic interactions between two populations. In vertebrates, motor rhythms arise from half-center oscillator modules in the spinal cord (*Marder and Calabrese, 1996*).

Although isolated CPGs can produce outputs in the absence of sensory input, in the intact animal sensory feedback plays a critical role in coordinating motor rhythms across the body and modulating their characteristics (*Friesen, 2009*; *Grillner and Wallén, 2002*; *Mullins et al., 2011*; *Pearson, 2004*; *Wen et al., 2012*). Sensory feedback allows animals to adapt locomotor patterns to their surroundings (*Andersson et al., 1981*; *Brodfuehrer and Friesen, 1986*) and adapt to unexpected perturbations (*Ekeberg and Grillner, 1999*). In leeches (*Cang et al., 2001*; *Cang and Friesen, 2000*) and *Drosophila* (*Akitake et al., 2015*; *Mendes et al., 2013*), specialized proprioceptive neurons and sensory receptors in body muscles detect sensory inputs to regulate and coordinate the centrally generated motor patterns. In limbed vertebrates, proprioceptors located in muscles, joints, and/or skin

detect body movements and interact with premotor interneurons to coordinate limb movements (*Pearson, 2004*). Sensory inputs induced by electric stimulation of receptor cells (*Yu and Friesen, 2004*) or by mechanical perturbation of body segments (*Grillner et al., 1981*) can entrain an animal's motor behavior to imposed patterns, demonstrating the flexibility of motor systems in responding to feedback.

Animal movements are driven not only by active muscle contractions but also by passive mechanical forces including elastic recoil of muscles and other body structures, internal damping forces, and forces from the interaction with the external environment. Efficient locomotion in vertebrates depends on storage of elastic energy in tendons and muscles (*Roberts and Azizi, 2011*). In insects, elasticity in the leg joint plays an important role in generating forces for walking and jumping (*Ache and Matheson, 2013*). A comprehensive understanding of animal locomotion should therefore encompass not only neural activity, muscle activity, and sensory feedback, but also biomechanical forces within the animal's body and between the animal and its environment (*Figure 1A*; *Borgmann et al., 2009*; *Grillner and Wallén, 2002*; *Kiehn, 1998*).

Here, we study mechanisms of locomotor rhythm generation and its modulation by sensory feedback in the nematode *Caenorhabditis elegans*. With its easily quantifiable behavior (*Croll, 1971*), well-mapped nervous system (*Cook et al., 2019*; *White et al., 1986*), genetic manipulability (*Bargmann, 1998*; *Brenner, 1974*; *Hobert, 2003*), and optical transparency, this worm is a unique model for obtaining an integrative understanding of locomotion.

*C. elegans* forward locomotion consists of anterior-to-posterior dorsoventral undulations (*Croll, 1971*). These movements are mediated by a neuromuscular circuit consisting of interneurons, excitatory cholinergic motor neurons, inhibitory GABAergic motor neurons, and body wall muscles. Laser ablation studies have shown that the cholinergic B-type motor neurons are required for forward locomotion (*Chalfie et al., 1985*). The GABAergic D-type motor neurons provide dorsoventral cross-inhibition to the body wall muscles and are essential for maintaining normal wave shape and frequency during forward locomotion (*Deng et al., 2021*; *McIntire et al., 1993*). A set of premotor interneurons (AVB, PVC, AVA, AVD, and AVE) regulate forward and reverse movements (*Chalfie et al., 1988*; *Driscoll and Kaplan, 1997*; *Von Stetina et al., 2006*). Ablation of all premotor interneurons or the D-type motor neurons does not deprive *C. elegans* of the ability to move forward (*Chalfie et al., 1985*; *Gao et al., 2018*; *Kawano et al., 2011*), suggesting that a network consisting of excitatory motor neurons and muscles may be sufficient to generate rhythmicity. Optogenetic and lesion experiments have shown that multiple oscillators exist in the ventral nerve

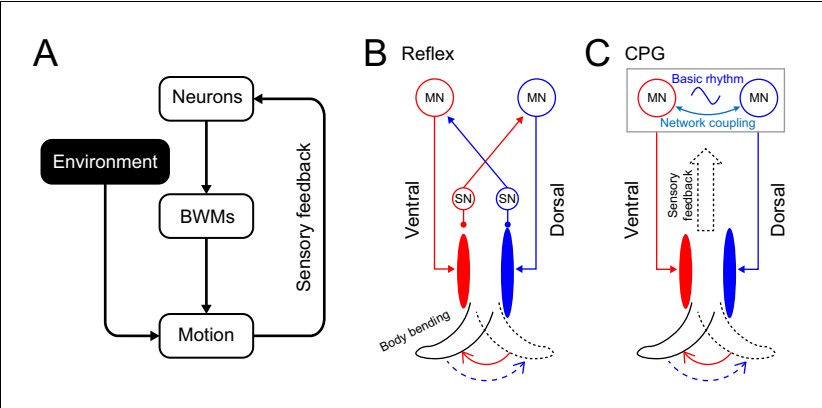

**Figure 1.** Rhythm generation in *C. elegans*. (**A**) Motor neurons generate neuronal signals to control the activation of body wall muscles (BWM), which generates movement subject to internal and external environmental constraints. Sensory input provides feedback about body position and the environment. (**B,C**) Two possible models for locomotory rhythm generation in *C. elegans*. (**B**) In a reflex loop model, sensory neurons (SN) detect body postures and excite motor neurons (MN) to activate body wall muscles. (**C**) In a central pattern generator (CPG) model, network of motor neurons generates basic rhythmic patterns that are transmitted to body wall muscles while sensory feedback modulates the CPG rhythm. Diagrams (**B-C**) are adapted from Figure 1 in *Marder and Bucher, 2001*.

cord (*Fouad et al., 2018*). However, the mechanisms that give rise to these oscillators are still poorly understood.

Proprioceptive feedback is crucial for *C. elegans* motor behavior. Studies have identified several neuron classes that have proprioceptive roles. The B-type motor neurons mediate proprioceptive coupling between anterior to posterior bending during forward locomotion (*Wen et al., 2012*). The SMDD motor neurons, localized at the head, have been identified as proprioceptive regulators of head steering during locomotion (*Yeon et al., 2018*). Both the B-type motor neurons and the SMDD head motor neurons have long asynaptic processes hypothesized to have proprioceptive function (*White et al., 1986*) and have been suggested as candidate locomotor CPG elements (*Kaplan et al., 2020*). In addition, two types of neurons, the DVA and PVD interneurons, have also been described as having proprioceptive roles in the regulation of worm's body bend movement. The cell DVA has been shown to exhibit proprioceptive properties with a dependence on a mechanosensitive channel, TRP-4, which acts as a stretch receptor to regulate the body bend amplitude during locomotion (*Li et al., 2006*). In another study, body bending was shown to induce local dendritic calcium transients in PVD and dendritic release of a neuropeptide encoded by *nlp-12*, which appears to regulate the amplitude of body movements (*Tao et al., 2019*).

To experimentally probe mechanisms of rhythmic motor generation, including the role of proprioceptive feedback, we measured the phase response curve (PRC) upon transient optogenetic inhibition of the head muscles. We found that the worms displayed a biphasic, sawtooth-shaped PRC with sharp transitions from phase delay to advance.

We used these findings to develop a computational model of rhythm generation in the *C. elegans* motor circuit in which a relaxation-oscillation process, with switching based on proprioceptive feedback, underlies the worm's rhythmic dorsal-ventral alternation. Computational models for *C. elegans* motor behavior have long been an important complement to experimental approaches, since an integrative understanding of locomotion requires consideration of neural, muscular, and mechanical degrees of freedom, and are often tractable only by modeling (*Boyle et al., 2012*; *Bryden and Cohen, 2008*; *Denham et al., 2018*; *Izquierdo and Beer, 2018*; *Johnson et al., 2021*; *Karbowski et al., 2008*; *Kunert et al., 2017*; *Olivares et al., 2021*). We sought to develop a phenomenological model to describe an overall mechanism of rhythm generation but not the detailed dynamics of specific circuit elements. We aimed to incorporate biomechanical constraints of the worm's body and its environment (*Fang-Yen et al., 2010*; *Gray and Lissmann, 1964*; *Wallace, 1968*), as well as account for how sensory feedback is incorporated. To improve predictive power, we aimed to minimize the number of free parameters used in the model. Finally, we sought to optimize and test this model with new experiments as well as with published findings.

Our model reproduces the observed PRC and describes the locomotory dynamics around optogenetic inhibitions in a manner that closely fits our experimental observations. Our model also agrees with results on gait adaptation to external load and the asymmetry in time-dependent curvature patterns of undulating worms. Our experimental findings and computational model together yield insights into how *C. elegans* generates rhythmic locomotion and modulates them depending on the environment.

## Results

### *C. elegans* forward locomotion exhibits a stable and nonsinusoidal limit cycle

To gain insight into wave generation, we first sought to examine the quantitative behavioral characteristics of worms during forward locomotion. First, we measured the undulatory dynamics of body bending by computing the time-varying curvature along the centerline of the body (*Fang-Yen et al., 2010*; *Leifer et al., 2011*; *Pierce-Shimomura et al., 2008*; *Wen et al., 2012*) from analysis of dark field image sequences of worms exhibiting forward locomotion. In order to quantitatively treat the drag between the body and its environment, we examined locomotion of worms in dextran solutions of known viscosity (see *Appendix*; *Fang-Yen et al., 2010*). The normalized body coordinate is defined by the distance along the body centerline divided by the body length (*Figure 2A*). The curvature $\kappa$ at each point along the centerline of the body is the reciprocal of local radius of curvature (*Figure 2A*), with a positive (negative) curvature representing ventral (dorsal) bending. We further

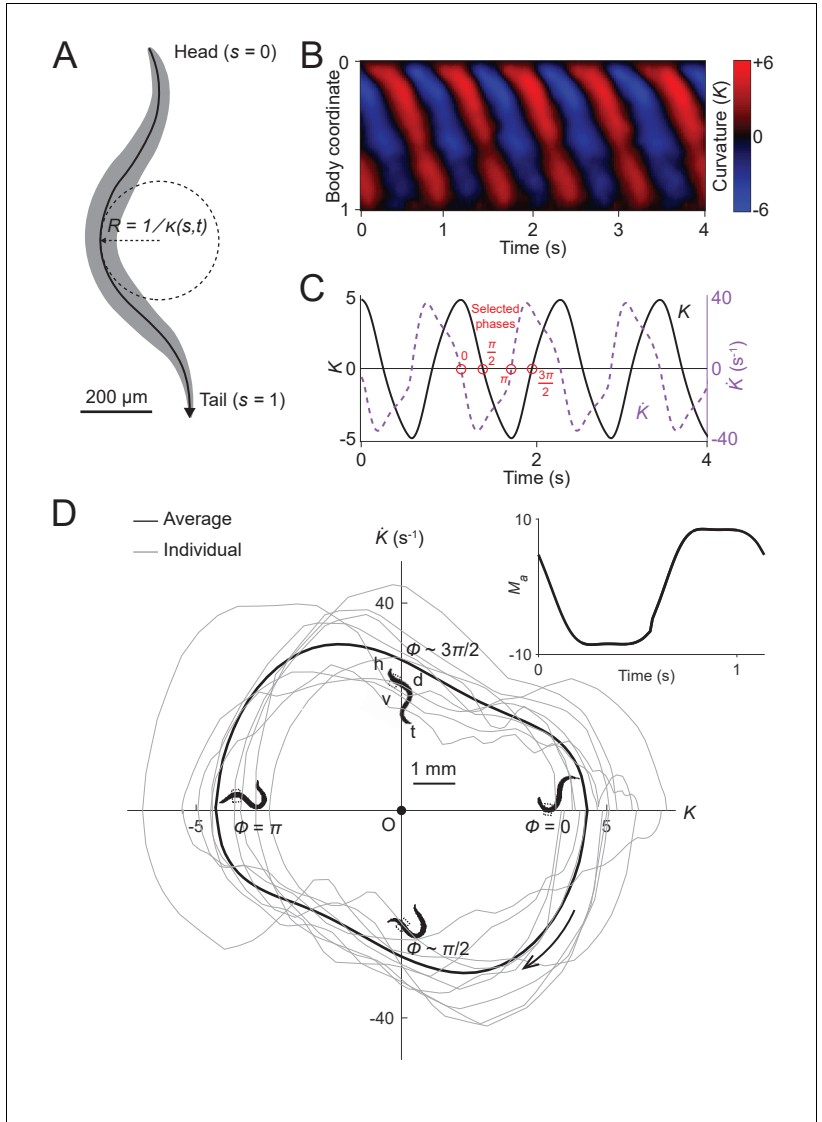

**Figure 2.** Undulatory dynamics of freely moving worms. (**A**) Worm undulatory dynamics are quantified by the time-varying curvature along the body. The normalized body coordinate is defined by the fractional distance along the centerline (head = 0, tail = 1). The curvature $\kappa$ is the reciprocal of the local radius of curvature with positive and negative values representing dorsal and ventral curvature, respectively. (**B**) Curvature as a function of time and body coordinate during forward movement in a viscous liquid. Body bending curvature $K$ is represented using the nondimensional product of $\kappa$ and body length $L$. (**C**) Curvature (black) in the anterior region (average over body coordinate 0.1-0.3) and the time derivative of curvature (dashed purple). Red circles mark four representative phases (0, $\pi/2$, $\pi$, and $3\pi/2$). The curve is an average of 5041 locomotory cycles from 116 worms. (**D**) Phase portrait representation of the oscillatory dynamics of the anterior region, showing the curvature and the time derivative of the curvature parameterized by time. Images of worm correspond to the phases marked in (**C**). Arrow indicates clockwise movement over time. Dash-boxed region of the worm body indicates the 0.1–0.3 body coordinates. h: head; t: tail; v: ventral side; d: dorsal side. Gray curves are individual locomotory cycles from freely moving worms (10 randomly selected cycles are shown). (*Inset*) waveform of the scaled active muscle moment, estimated by equation $M_a = K + \tau_u \dot{K}$. Both curves were computed from the data used in (**C**).

The online version of this article includes the following figure supplement(s) for figure 2:

**Figure supplement 1.** Phase portrait representations of the oscillatory bending dynamics for various body coordinates.

define the dimensionless or scaled curvature $K = \kappa \cdot L$, where $L$ is the length of the worm. Using this metric, we quantified the worm's forward movement by calculating scaled curvature as a function of body coordinate and time (*Figure 2B*).

We used this behavioral data to generate phase portraits, geometric representations of a dynamical system's trajectories over time (*Izhikevich, 2007*), in which the time derivative of the curvature is plotted against the curvature. If the curvature were sinusoidal over time, as it is often modeled in slender swimmers (*Fang-Yen et al., 2010*; *Gray, 1933*; *Guo and Mahadevan, 2008*; *Niebur and Erdös, 1991*), the time derivative of curvature would also be sinusoidal, with a phase shift of $\pi/4$ radians relative to the curvature, and the resulting phase portrait would be symmetric about both the $K$ and $dK/dt$ axes. Instead, we found that the phase portrait of the bending movement in the worm's head region (0.1–0.3 body coordinate) during forward locomotion is in fact non-ellipsoidal and strongly asymmetric with respect to reflection across the $K$ or $dK/dt$ axes (*Figure 2D*). Plots of both the phase portrait (*Figure 2D*) and the time dependence (*Figure 2C*) show that $K$ and $dK/dt$ are strongly non-sinusoidal.

In addition to the head, other parts of the worm's body also display nonsinusoidal bending movements (*Figure 2—figure supplement 1*). In this paper, we focus on curvature dynamics of the worm's head region (0.1–0.3 body coordinate) where the bending amplitude is largest and the non-sinusoidal features are most prominent (*Figure 2—figure supplement 1*).

We asked whether the phase portrait represents a stable cycle, that is whether the system tends to return to the cycle after fluctuations or perturbations away from it. To this end, we analyzed the recovery after brief optogenetic muscle inhibition. We used a closed-loop system for optically targeting specific parts of the worm (*Fouad et al., 2018*; *Leifer et al., 2011*) to apply brief pulses of laser illumination (0.1 s duration, 532 nm wavelength) to the heads of worms expressing the inhibitory opsin *NpHR* in body wall muscles (via the transgene *Pmyo-3::NpHR*). Simultaneous muscle inhibition on both sides causes *C. elegans* to straighten due to internal elastic forces (*Fang-Yen et al., 2010*). Brief inhibition of the head muscles during forward locomotion was followed by a maximum degree of paralysis approximately 0.3 s after the end of the pulse, then a resumption of undulation (*Figure 3A,B*; *Video 1*).

To quantify the recovery dynamics, we defined a normalized deviation $d$ describing the state of the system relative to the phase portrait of normal oscillation (see *Appendix*), such that $d = -1$ at the origin, $d = 0$ at the limit cycle, and $d > 0$ outside the limit cycle. We found that the deviation following optogenetic perturbation (*Figure 3—figure supplement 1*) decays toward zero regardless of the initial deviation from the normal cycle, indicating that the worm tends to return to its normal oscillation after a perturbation. These results show that *C. elegans* head oscillation during forward locomotion is stable under optogenetic perturbation. The dynamics of these perturbed worms also allow us to reconstruct the phase isochrons and vector flow fields (*Figure 3—figure supplement 2*) of the worm's head oscillation, two other important aspects of an oscillator (see *Appendix*).

Taken together, these results show that during forward locomotion, head oscillation of a worm constitutes a stable oscillator containing a nonsinusoidal limit cycle.

## Transient optogenetic inhibition of head muscles yields a slowly rising, rapidly falling phase response curve

The phase response curve (PRC) describes the change in phase of an oscillation induced by a perturbation as a function of the phase at which the perturbation is applied, and is often used to characterize biological and nonbiological oscillators (*Izhikevich, 2007*; *Pietras and Daffertshofer, 2019*; *Schultheiss et al., 2011*). We performed a phase response analysis of the worm's locomotion upon transient optogenetic inhibitions.

Using data from 991 illuminations (each 0.1 s in duration) in 337 worms, we analyzed the animals' recovery from transient paralysis as a function of the phase at which the illumination occurred. We define the phase such that it equals to zero at the point of maximum ventral bending (*Figure 3D*). When inhibition occurred with phase in the interval $[0, \pi/6]$, the head typically straightened briefly and then continued the previous bend, resulting in a phase delay for the oscillation (*Figure 3C–E*). When inhibition occurred with phase in the interval $[\pi/3, \pi/2]$, the head usually appeared to discontinue the previous bend movement, which resulted in a small phase advance (*Figure 3F–H*). When inhibition occurred with phase in the interval $[2\pi/3, 5\pi/6]$, the head response was similar to that within the interval $[0, \pi/6]$, and also resulted in a phase delay (*Figure 3I–K*).

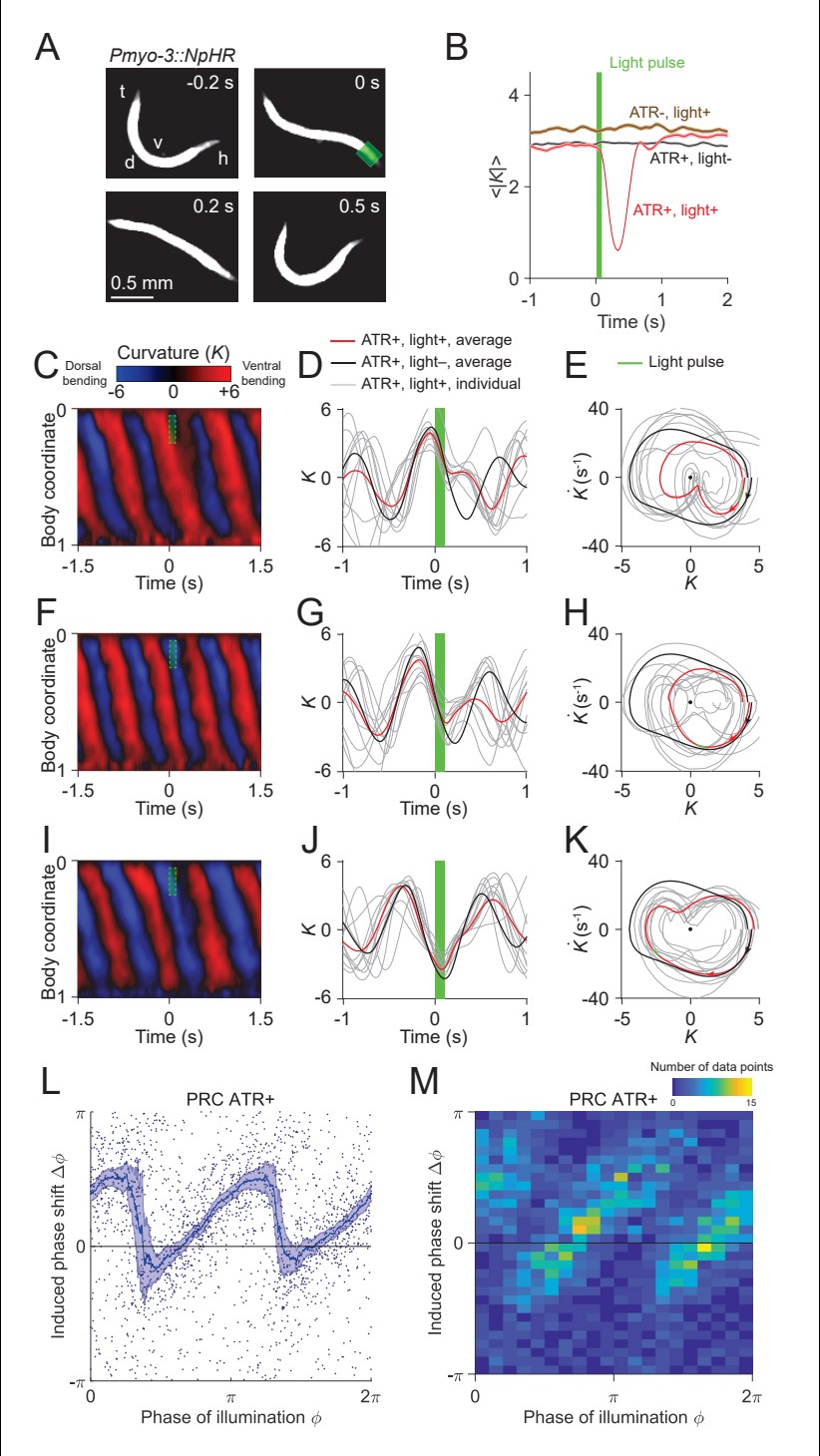

**Figure 3.** Analysis of phase-dependent inhibitions for head oscillation using transient optogenetic muscle inhibition. (**A**) Images of a transgenic worm (*Pmyo-3::NpHR*) perturbed by a transient optogenetic muscle inhibition in the head during forward locomotion. Green shaded region indicates the 0.1 s laser illumination interval. h: head; t: tail; v: ventral side; d: dorsal side. (**B**) Effect of muscle inhibition on mean absolute curvature of the head. Black curve represents control ATR+ (no light) group (3523 measurements using 337 worms). Brown curve represents control ATR- group (2072 measurements using 116 worms). Red curve represents ATR+ group (1910 measurements using 337 worms). Green bar indicates 0.1 s light illumination interval starting at $t = 0$. (**C-E**) Perturbed dynamics around light pulses occurring in the phase range $[0, \pi/6]$. (**C**) Kymogram of time-varying curvature $K$ around a 0.1 s inhibition (green dashed box). (**D**) Mean curvature dynamics around the inhibitions

*Figure 3 continued on next page*

*Figure 3 continued*

(green bar, aligned at $t = 0$) from ATR+ group (red curve, 11 trials using 4 worms) and control ATR+ (no light) group (black curve, eight trials using three worms). Gray curves are individual trials from ATR+ group (10 randomly selected trials are shown). (**E**) Mean phase portrait graphs around the inhibitions (green line) from ATR+ group (same trials as in **D**) and control group (ATR+, no light, 3998 trials using 337 worms). Gray curves are individual trials from ATR+ group. (**F-H**) Similar to (**C-E**), for phase range $[\pi/3, \pi/2]$. (**I-K**) Similar to (**C-E**), for phase range $[2\pi/3, 5\pi/6]$. (**L**) PRC from optogenetic inhibition experiments (ATR+ group, 991 trials using 337 worms, each point indicating a single illumination of one worm). The curve was obtained via a moving average along the *x*-axis with $0.16\pi$ in bin width and the filled area represents 95% confidence interval within the bin. (**M**) A 2-dimensional histogram representation of the PRC using the same data. The histogram uses 25 bins for both dimensions, and the color indicates the number of data points within each rectangular bin.

The online version of this article includes the following figure supplement(s) for figure 3:

**Figure supplement 1.** Normalized deviation to the normal cycle (the unperturbed oscillation) for the head oscillation of the perturbed worms.
**Figure supplement 2.** The isochron map overlaid with the vector field for the worm's head oscillation.
**Figure supplement 3.** Phase response curve of *Pmyo-3::NpHR* worms (ATR- control group).
**Figure supplement 4.** Phase response curve of *Pmyo-3::NpHR* worms perturbed by a 0.055 s optogenetic muscle inhibition during normal locomotion.
**Figure supplement 5.** Phase response curves of *Pmyo-3::NpHR* worms induced by a 0.1 s optogenetic muscle inhibition, perturbed and measured at various body regions.
**Figure supplement 6.** Phase response curve of transgenic worms that express *NpHR* in all cholinergic neurons (*Punc-17::NpHR::ECFP*).
**Figure supplement 7.** Phase response curve of transgenic worms that express Arch in the B-type motor neurons (*Pacr-5::Arch-mCherry*).
**Figure supplement 8.** Phase response curve of transgenic worms that express *NpHR* in the body wall muscles but lack the GABA receptor for the D-type motor neurons (*Pmyo-3::NpHR; unc-49(e407)*).

Combining the data from all phases of inhibition yielded a sawtooth-shaped PRC with two sharp transitions from phase delay to advance as well as two relatively slow ascending transitions from phase advance to delay (*Figure 3L,M*). In control worms, which do not express *NpHR* in the body wall muscles (see Materials and methods), the resulting PRC shows no significant phase shift over any phases of illumination (*Figure 3—figure supplement 3*). In worms perturbed with shorter pulses (0.055 s duration), we observed a similar sawtooth-shaped PRC (*Figure 3—figure supplement 4*).

In addition to phase response analyses with perturbations to the worm's anterior region, we conducted similar analyses for the dynamics across the body by optogenetically inhibiting body wall muscles of other regions (*Figure 3—figure supplement 5*). We found that the sawtooth feature of PRC tends to decrease monotonically as the perturbation occurs further away from the head (*Figure 3—figure supplement 5A,E,I*).

Next, we asked whether the sharp downward transitions in the PRC represent a continuous decrease or instead result from averaging data from a bimodal distribution. When we plotted the distribution of the same data in a 2-D representation we found that the phase shifts display a piecewise, linear increasing dependence on the phase of inhibition with two abrupt jumps occurring at $\phi \approx \pi/3$ and $4\pi/3$, respectively (*Figure 3M*). This result shows that the sharp decreasing transitions in PRC reflect bimodality in the data rather than continuous transitions.

In addition to examining PRCs induced by muscle inhibition, we also calculated PRCs with respect to inhibitions of cholinergic motor neurons. We performed similar experiments on transgenic worms in which the inhibitory opsin NpHR is expressed in either all cholinergic neurons (*Punc-17::NpHR::ECFP*) or B-type motor neurons (*Pacr-5::Arch-mCherry*). In both strains,

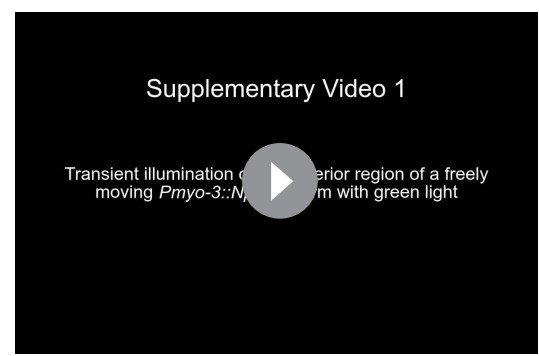

**Video 1.** Transient illumination of the anterior region of a freely moving *Pmyo-3::NpHR* worm. Green-shaded region indicates timing and location of illumination. https://elifesciences.org/articles/69905#video1

we again observed sawtooth-shaped PRCs (*Figure 3—figure supplements 6* and *7*), with variations only in the magnitudes of phase shifts. These experiments show that the sawtooth-shaped feature of PRC is maintained for motor neuron inhibition, suggesting that the transient muscle and neuron inhibition interrupt the motor circuit dynamics in a similar manner.

The GABAergic D-type motor neurons provide a dorsoventral reciprocal inhibition of opposing muscles during locomotion. We asked whether the D-type motor neurons are required for the observed sawtooth shape of the PRC. We examined transgenic worms that express *NpHR* in the body wall muscles but have mutations *unc-49(e407)*, a loss-of-function mutant of GABA$_A$ receptor that is required by the D-type motor neurons (*Bamber et al., 1999*). After performing optogenetic inhibition experiments, we found that the PRC also displays a sawtooth feature (*Figure 3—figure supplement 8*). This result shows that D-type motor neurons are not necessary for the motor rhythm generator to show the sawtooth-shaped PRC.

Sawtooth-shaped PRCs are observed in a number of systems with oscillatory dynamics, including the van der Pol oscillator (*Cestnik and Rosenblum, 2018*), and may reflect a phase resetting property of an oscillator with respect to a perturbation (*Izhikevich, 2007*; *Schultheiss et al., 2011*). Further interpretation of the PRC results is given below.

## Worm muscles display a rapid switch-like alternation during locomotion

As a first step in interpreting and modeling our findings, we estimated the patterns of muscle activity in freely moving worms, in part by drawing on previous biomechanical analyses of nematode movement (*Fang-Yen et al., 2010*; *Gray and Lissmann, 1964*; *Wallace, 1968*).

In mechanics, a moment is a measure of the ability of forces to produce bending about an axis. Body wall muscles create local dorsal or ventral bending by generating active moments across the body. In addition to the active moments from muscles, there are also passive moments generated by the worm's internal viscoelasticity and by the forces due to the interaction of the worm with its external environment.

We estimated the output patterns of the active muscle moment that drives the head oscillations of freely moving worms immersed in viscous solutions. Following previous analyses of *C. elegans* locomotor biomechanics under similar external conditions (*Fang-Yen et al., 2010*), the scaled active muscle moment can be described as a linear combination of the curvature and the time derivative of the curvature (*Equation 1*; also see *Methods* and *Appendix*). We observed that in the phase portrait graph (*Figure 2D*), there are two nearly linear portions of the curve. We hypothesized that these linear portions correspond to two bouts during which the active muscle moment is nearly constant.

Using fits to the phase plot trajectory (see Materials and methods and *Appendix*) we estimated the waveform of the active muscle moment as a function of time (*Figure 2D Inset*). We found that the net active muscle moment alternates between two plateau regions during forward locomotion. From the slope of the steep portions on this curve, we estimated the time constant for transitions between active moments to be $\tau_m \approx 100\,ms$. This time constant is much smaller than the duration of each muscle moment plateau period ($\approx 0.5\,s$), suggesting that the system undergoes rapid switches of muscle contractions between two saturation states.

## A relaxation oscillator model explains nonsinusoidal dynamics

We reasoned that the rapid transitions of the active muscle moment might reflect a switching mechanism in the locomotory rhythm generation system. We hypothesized that the motor system generates locomotory rhythms by switching the active moment of the muscles based on proprioceptive thresholds.

To expand further upon these ideas, we developed a quantitative model of locomotory rhythm generation. We consider the worm as a viscoelastic rod where the scaled curvature *K(t)* varies according to:

$$K(t) + \tau_u \frac{dK(t)}{dt} = M_a(t), \tag{1}$$

where $\tau_u$ describes the time scale of bending relaxation and $M_a(t)$ is the time-varying active muscle moment scaled by the bending modulus and the body length (see detailed derivations in *Appendix*). We note that in a stationary state ($dK/dt = 0$), the curvature would be equal to the scaled active

muscle moment. That is, the scaled active moment represents the static curvature that would result from a constant muscle moment.

We define a proprioceptive feedback variable $P$ as a linear combination of the current curvature value and the rate of change of curvature. In our model, once this variable reaches either of two thresholds $P_{th}$ and $-P_{th}$ (*Figure 4D*), the active muscle moment undergoes a change of sign (*Figure 4E*), causing the head to bend toward the opposite direction (*Figure 4B*).

Our model has 5 parameters: (1) $\tau_u$, the bending relaxation time scale, (2) $\tau_m$, the muscle switching time scale, (3) $M_0$, the amplitude of the scaled active muscle moment, (4-5) $b$ and $P_{th}$, which determine the switch threshold. The first three parameters were directly estimated from our experimental results from freely moving worms (see *Appendix*). Parameters $b$ and $P_{th}$ were obtained using a two-round fitting procedure by fitting the model first to the freely moving dynamics (first round) and then to the experimental phase response curve (second round) (see *Appendix*).

With this set of parameters, we calculated the model dynamics as represented by the phase portrait (*Figure 4C*) as well as curvature waveform in one cycle period (*Figure 4F*). We found that in both cases the model result agreed with our experimental observations. Our model captures the asymmetric phase portrait trajectory shape found from our experiments (*Figure 2D*). It also describes the asymmetry of head bending during locomotion: bending toward the ventral or dorsal directions occurs slower than straightening toward a straight posture during the locomotory cycle (*Figure 4F* Inset).

Considering the hypothesized mechanism under the biomechanical background (*Equation 1*), our model provides a simple explanation for the observed bending asymmetry during locomotion. According to the model, the active muscle moment is nearly constant during each period between transitions of the muscle moment. Biomechanical analysis under this condition predicts an approximately exponential decay in curvature, which gives rise to an asymmetric feature during each half period (*Figure 4F*).

## Relaxation oscillator model reproduces responses to transient optogenetic inhibition

We performed simulations of optogenetic inhibitions in our model. To model the transient muscle paralysis, the muscle moment is modulated by a bell-shaped function of time (*Figure 4—figure supplement 1*; also see *Appendix*) such that, upon inhibition, it decays toward zero and then recovers to its normal value, consistent with our behavioral observations (*Figure 3B*).

From simulations with different sets of model parameters, we found that the model PRCs consistently exhibited the sawtooth shape found in experiments, although differing in height and timing of the downward transitions. In addition to the model parameters $\tau_u$, $M_0$, and $\tau_m$ that had been explicitly estimated from free-moving experiments, we performed a two-round fitting procedure (see *Appendix*) to determine the other parameters (including $b$, $P_{th}$, and parameters for describing the optogenetically induced muscle inhibitions (see *Figure 4—figure supplement 1*)) to best fit the freely moving dynamics and the experimental PRC, respectively, with a minimum mean squared error (MSE) (*Figures 4F* and *5A*; also see *Appendix*). For the parameters $b$ and $P_{th}$, the optimization estimated their values to be $b = 0.046\,s$ and $P_{th} = 2.33$, as shown on the phase portraits (gray dashed lines in *Figures 4C*, *5B and D*).

The threshold-switch mechanism model provides an explanation for the observed sawtooth-shaped PRC. By comparing model phase portrait graphs around inhibitions occurring at different phases (*Figure 5B–E*), we found that the phase shift depends on the relative position of the inhibition with respect to the switch points on the phase plane. (1) If the effect of the inhibition occurs before the system reaches its switch point (*Figure 5B*), the system will recover by continuing the previous bend and the next switch in the muscle moment will be postponed, thereby leading to a phase delay (*Figure 5C*). (2) As the inhibition progressively approaches the switch point, one would expect that the next switch in the muscle moment will also be progressively postponed; this explains the increasing portions of the PRC. (3) If the inhibition coincides with the switch point (*Figure 5D*), the muscle moment will be switched at this point and the system will recover by aborting the previous bend tendency, resulting in a small phase advance (*Figure 5E*). This switching behavior explains the two sharp downward transitions in the PRC.

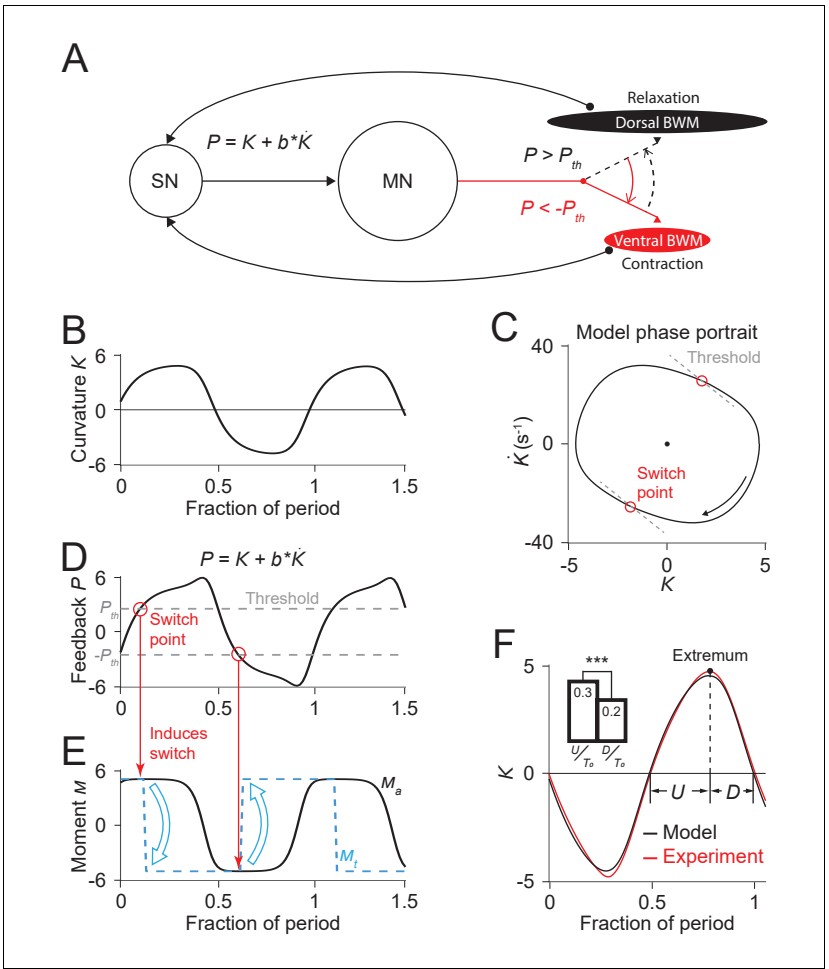

**Figure 4.** Free-running dynamics of a bidirectional relaxation oscillator model. (A) Schematic diagram of the relaxation oscillator model. In this model, sensory neurons (SN) detect the total curvature of the body segment as well as the time derivative of the curvature. The linear combination of the two values, $P = K + b\dot{K}$, is modeled as the proprioceptive signal which is transmitted to motor neurons (MN). The motor neurons alternately activate dorsal or ventral body wall muscles (BWM) based on a thresholding rule: (1) if $P < -P_{th}$, the ventral body wall muscles get activated and contract while the dorsal side of muscles relax; (2) if $P > P_{th}$, vice versa. Hence, locomotion rhythms are generated from this threshold-switch process. (B) Time-varying curvature $K$ of the model oscillator. The time axis is normalized with respect to oscillatory period (same for D, E, and F). (C) Phase portrait graph of the model oscillator. Proprioceptive threshold lines (gray dashed lines) intersect with the phase portrait graph at two switch points (red circles) at which the active moment of body wall muscles is switched. (D) Time-varying proprioceptive feedback $P$ received by the motor neurons. Horizontal lines denote the proprioceptive thresholds (gray dashed lines) that switch the active muscle moment at switch points (red circles, intersections between the proprioceptive feedback curve and the threshold lines). (E) Time-varying active muscle moment. Blue-dashed square wave denotes target moment ($M_t$) that instantly switches directions at switch points. Black curve denotes the active muscle moment ($M_a$) which follows the target moment in a delayed manner. (F) Time varying curvature in the worm's head region from experiments (red, 5047 cycles using 116 worms) and model (black). Model curvature matches experimental curvature with an MSE ≈ 0.18. (Inset) Bar graph of $U$ (time period of bending toward the ventral or dorsal directions) and $D$ (time period of straightening toward a straight posture). Vertical bars are averages of fractions with respect to undulatory period $T_0$ of $U$ and $T$ (*** indicates $p < 0.0005$ using Student's $t$ test).

The online version of this article includes the following figure supplement(s) for figure 4:

**Figure supplement 1.** Bell-shaped function for modeling the optogenetic muscle inhibition (*Equation A14*).

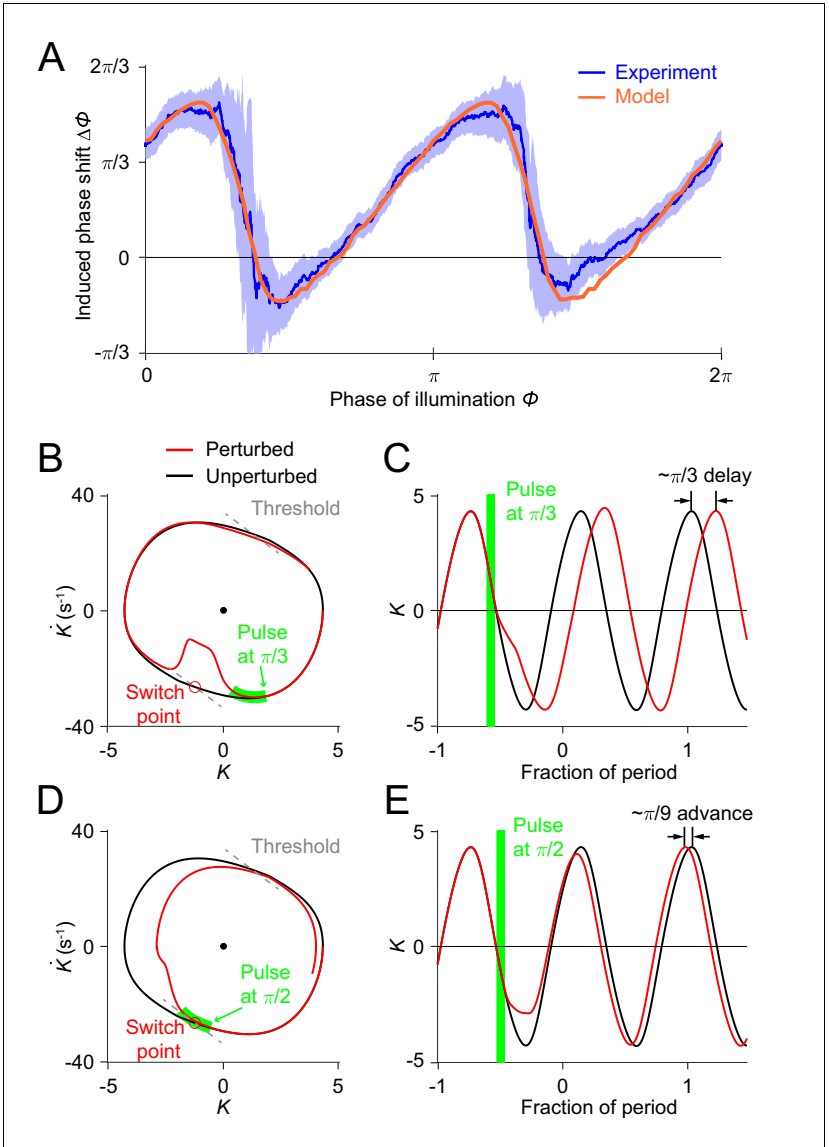

**Figure 5.** Simulations of optogenetic inhibitions in the relaxation oscillator model. (**A**) Phase response curves measured from experiments (blue, same as in *Figure 3L*) and model (orange). Model PRC matches experimental PRC with an MSE ≈ 0.12. (**B,C**) Simulated dynamics of locomotion showing inhibition-induced phase delays in the model oscillator. (**B**) Simulated phase portrait graphs around inhibition occurring at $\pi/6$ phase of cycle for perturbed (red) and unperturbed (black) dynamics. Green bar indicates the phase during which the inhibition occurs. (**C**) Same dynamics as in (**B**), represented by time-varying curvatures. The time axis is normalized with respect to oscillatory period (same for **E**). (**D,E**) Simulated dynamics of locomotion showing inhibition-induced phase advances in the model oscillator. (**D**) Simulated phase portrait graphs around inhibition occurring at $\pi/2$ phase of cycle for perturbed (red) and unperturbed (black) dynamics. (**E**) Same dynamics as in (**D**), represented by time-varying curvatures.

The online version of this article includes the following figure supplement(s) for figure 5:

**Figure supplement 1.** Performance of model oscillators: threshold-switch (column 1), van der Pol (column 2), Rayleigh (column 3), and Stuart-Landau (column 4).

## Relaxation oscillator model predicts phase response curves for single-side muscle inhibition

As a further test of the model, we asked what PRCs would be produced with only the ventral or dorsal head muscles being transiently inhibited. In the model, the muscle activity is represented using

the scaled active moment of muscles. We conducted model simulations (see *Appendix*) to predict the PRCs for transient inhibitions of muscles on the dorsal side (*Figure 6A*, *Upper*) and ventral side (*Figure 6B*, *Upper*), respectively.

To experimentally perform phase response analysis of single-side muscle inhibitions, we visually distinguished each worm's dorsoventral orientation (via vulval location) and targeted light to either the ventral or dorsal side of the animal. Transiently illuminating (0.1 s duration) dorsal or ventral muscles in the head region of the transgenic worms (*Pmyo-3::NpHR*) induced a brief paralyzing effect when the segment was bending toward the illuminated side but did not induce a significant paralyzing effect when the segment was bending away from the illuminated side (*Figure 6—figure supplement 1*).

Combining the experimental data from all phases of dorsal-side or ventral-side inhibition yielded the corresponding PRCs (*Figure 6A,B*, respectively), from which we found that both PRCs show a peak in the phase range during which the bending side is illuminated but shows no significant phase shift in the other phase range. The experimental observations are qualitatively consistent with model predictions.

We found that the PRC of dorsal-side illumination shows a smaller paralytic response than that of ventral-side illumination. This discrepancy may be due to different degrees of paralysis achieved during ventral vs. dorsal illumination (*Figure 6—figure supplement 1*), possibly due to differences in levels of opsin expression and/or membrane localization. We therefore modulated the parameter for describing degree of paralysis when simulating the PRC of the dorsal-side illumination to qualitatively account for this discrepancy (see *Appendix*).

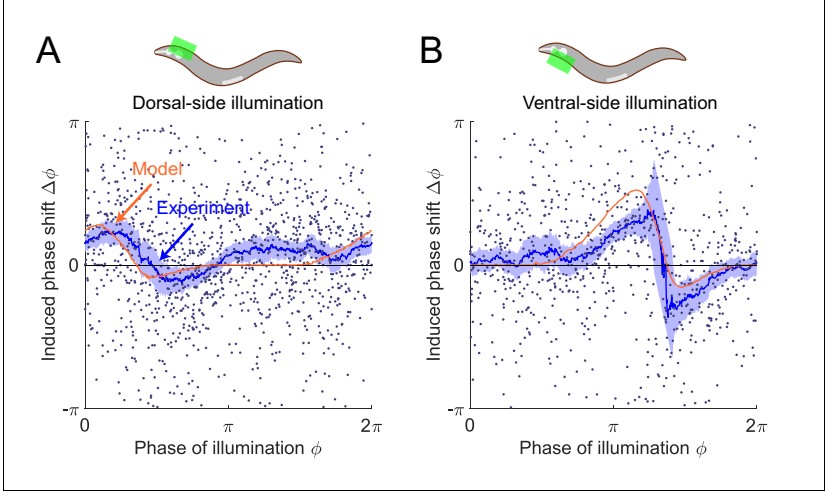

**Figure 6.** The model predicts phase response curves with respect to single-side muscle inhibitions. (**A**) (*Upper*) a schematic indicating a transient inhibition of body wall muscles of the head on the dorsal side. (*Lower*) the corresponding PRC measured from experiments (blue, 576 trials using 242 worms) and model (orange). (**B**) (*Upper*) a schematic indicating a transient inhibition of body wall muscles of the head on the ventral side. (*Lower*) the corresponding PRC measured from experiments (blue, 373 trials using 176 worms) and model (orange). For the two experiments, each point indicates a single illumination (0.1 s duration, 532 nm wavelength) of one worm. Experimental curves were obtained using a moving average along the x-axis with $0.16\pi$ in bin width. Filled area of each experimental curve represents 95% confidence interval with respect to each bin of data points.
The online version of this article includes the following figure supplement(s) for figure 6:

**Figure supplement 1.** Paralyzing effect analysis of muscle inhibitions induced by illumination on different sides of the worm's head segment.

**Figure supplement 2.** Phase response curves with respect to single-side muscle inhibition, simulated from model oscillators: threshold-switch (column 1), van der Pol (column 2), Rayleigh (column 3), and Stuart-Landau (column 4).

## Our model is consistent with the dependence of wave amplitude and frequency on external load

*C. elegans* can swim in water and crawl on moist surfaces, exhibiting different undulatory gaits characterized by different frequency, amplitude, and wavelength (*Figure 7A*). Previous studies *Berri et al., 2009*; *Fang-Yen et al., 2010* have shown that increasing viscosity of the medium induces a continuous transition from a swimming gait to a crawling gait, characterized by a decreasing undulatory frequency (*Figure 7C*) and an increasing curvature amplitude (*Figure 7D*). We asked whether our model is consistent with this load-dependent gait adaptation.

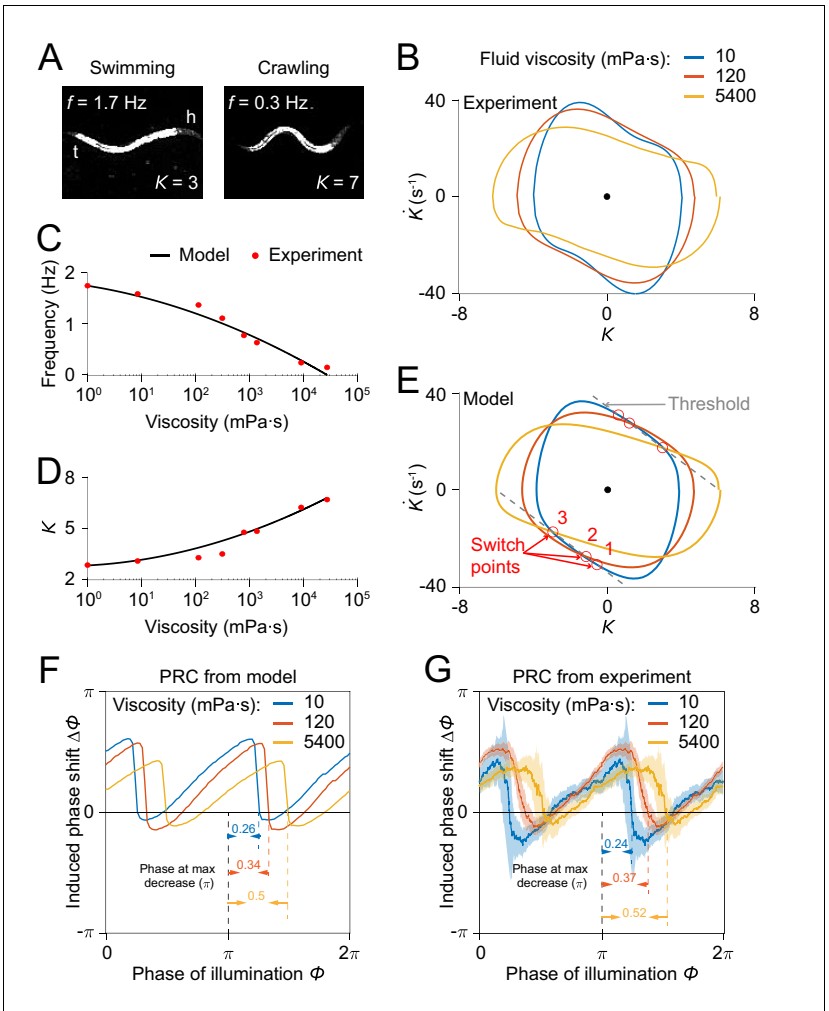

**Figure 7.** Model reproduces *C.elegans* gait adaptation to external viscosity. (**A**) Dark field images and the corresponding undulatory frequencies and amplitudes of adult worms (*left*) swimming in NGM buffer of viscosity 1 mPa·s, (*right*) crawling on agar gel surface. The worm head is to the right in both images. (**B**) Phase portrait graphs measured from worm forward movements in fluids of viscosity 10 mPa·s (blue, 3528 cycles using 50 worms), 120 mPa·s (red, 5050 cycles using 116 worms), and 5400 mPa·s (yellow, 1364 cycles using 70 worms). (**C,D**) The model predicts the dependence of undulatory frequency (**C**) and curvature amplitude (**D**) on external viscosity (black) that closely fit the corresponding experimental observations (red). (**E**) Phase portrait graphs predicted from the model in three different viscosities (same values as in **B**). Gray dashed lines indicate threshold lines for dorsoventral bending. The intersections (red circles 1, 2, 3) between the threshold line and phase portrait graphs are switch points for undulations in low, medium, high viscosity, respectively. (**F**) Theoretically predicted PRCs in fluids of the three different viscosities show that PRC will be shifted to the right as the viscosity of environment increases. (**G**) PRCs measured from optogenetic inhibition experiments in the three viscosities. Experimental PRCs were obtained using a moving average along the *x*-axis with $0.16\pi$ in bin width and filled areas are 95% confidence interval. The tendency of shift observed in experimental PRCs verified the model prediction.

We incorporated the effect of external viscosity into our model through the bending relaxation time constant $\tau_u$ (see *Appendix*). We ran our model to determine the dependence of model output on viscosity with varying viscosity $\eta$. We found that model results for frequency and amplitude dependence on viscosity of the external medium are in quantitative agreement with previous experimental results (*Fang-Yen et al., 2010*; *Figure 7C,D*).

We sought to develop an intuitive understanding of how the model output changes with increasing viscosity. We recall that the model generates a proprioceptive feedback variable in the form $P = K + b\dot{K}$ (*Figure 4A*), and that the active muscle moment in our model undergoes a change of sign upon the proprioceptive feedback reaching either of two thresholds, $P_{th}$ and $-P_{th}$. As the viscosity increases, one expects that a worm will perform a slower undulation due to the increase in external load. That is, the term $b\dot{K}$ becomes smaller. To compensate for this effect, the worm needs to undulate with a larger curvature amplitude to maintain the same level of proprioceptive feedback.

Next, we asked how the PRC depends on external viscosity. Model simulations with three different viscosities produced PRCs with similar sawtooth shape but with sharp transitions delayed in phase as the external viscosity increases (*Figure 7F*). We also measured PRCs from optogenetic inhibition experiments in solutions of three different viscosities (*Figure 7G*). Comparing the relative locations of the transitions in PRCs between the model and the data, our prediction also quantitatively agrees with the experimental results.

These results further support the model's description of how undulatory dynamics are modulated by the external environment.

## Evaluation of alternative oscillator models

Although our computational model agrees well with our experimental results, we asked whether other models could also explain our findings. We examined three alternative models based on well-known mathematical descriptions of oscillators (van der Pol, Rayleigh, and Stuart-Landau oscillators) and compared them with our original threshold-switch model and with our experimental data.

First, we tested the van der Pol oscillator, the first relaxation oscillator model (*Van der Pol, 1926*) which has long been applied in modeling neuronal dynamics (*Fitzhugh, 1961*; *Nagumo et al., 1962*). It is based on a second-order differential equation for a harmonic oscillator with a nonlinear, displacement-dependent damping term (see *Appendix*). By choosing a set of appropriate parameters, we found that the free-running waveform and phase plot of the van der Pol oscillator are highly asymmetric, but in an inverted manner (*Figure 5—figure supplement 1B,F*), compared with the experimental observations (*Figure 2C,D*). Transiently perturbing the system with the bell-shaped modulatory function over all phases within a cycle produced a similar sawtooth-shaped PRC as that observed experimentally (*Figure 5—figure supplement 1N*). However, the perturbed system was found to recover toward its limit cycle with a much slower rate than that of the experiments (*Figure 5—figure supplement 1J*). Simulations of single-side muscle inhibitions to the system produced single-sawtooth-shaped PRCs similar to those found experimentally (*Figure 6—figure supplement 2B,F*).

Next, we examined the Rayleigh oscillator, another relaxation oscillator model which was originally proposed to describe self-sustained acoustic vibrations such as vibrating clarinet reeds (*Rayleigh, 1896*). It is based on a second-order differential equation with a nonlinear, velocity-dependent damping term and it can be obtained from the van der Pol oscillator via a variable differentiation and substitution (see *Appendix*). From its free-running dynamics, we observed that the system exhibits a highly asymmetric waveform and phase plot that are similar to the experimental observations (*Figure 5—figure supplement 1C,G*). Additionally, the Rayleigh oscillator also produces similar sawtooth-shaped PRCs with respect to transient muscle inhibitions of both sides (*Figure 5—figure supplement 1O*), dorsal side (*Figure 6—figure supplement 2C*), and ventral side (*Figure 6—figure supplement 2G*), respectively, and system's recovery rate after the perturbation was shown to be similar to that of the experiments (*Figure 5—figure supplement 1K*).

Finally, we considered the Stuart-Landau oscillator, a commonly used model for the analysis of neuronal synchrony (*Acebrón et al., 2005*). Its nonlinearity is based on a negative damping term which depends on the magnitude of the state variable defined in a complex domain (see *Appendix*). The negative damping of the system constantly neutralize the positive damping on a limit cycle, making its free-running dynamics a harmonic oscillation which shows a sinusoidal waveform

(*Figure 5—figure supplement 1D,H*). Moreover, PRCs with respect to transient muscle inhibitions are constant with respect to phase (*Figure 5—figure supplement 1P*), contrary to the experiments.

We compared the results of our models with the experimental results. In the van der Pol oscillator, the free-running waveform displays a different asymmetry (*Figure 5—figure supplement 1B,F*) compared with the experimental observations and the perturbed system was shown to recover toward its limit cycle with a much slower rate than that of the experiments (*Figure 5—figure supplement 1J*). The Rayleigh oscillator reproduces a free-running waveform similar to experimental ones (*Figure 5—figure supplement 1C,G*) and its recovery rate toward limit cycle upon perturbation was close to that of the experiments (*Figure 5—figure supplement 1K*). However, its PRC (*Figure 5—figure supplement 1O*) showed weaker agreement with the experimental PRC compared with the threshold-switch model (*Figure 5—figure supplement 1M*) or the van der Pol model (*Figure 5—figure supplement 1N*). Of all the models tested, the threshold-switch model showed the least mean-square error with the PRC data (*Figure 5—figure supplement 1M–P*). We conclude that of these models, our threshold-switch model produced the best overall agreement with experiments.

We also found that two important experimental findings, the nonsinusoidal free-moving dynamics and the sawtooth-shaped PRCs can be achieved in our original model, the van der Pol and Rayleigh oscillators, which are all relaxation oscillators, but not in the Stuart-Landau oscillator, which is not a relaxation oscillator. Taken together, these results are consistent with the idea that a relaxation oscillation mechanism may underlie *C. elegans* motor rhythm generation.

## Discussion

In this study, we used a combination of experimental and modeling approaches to probe the mechanisms underlying the *C. elegans* motor rhythm generation.

Our model can be compared to those previously described for *C. elegans* locomotion. An early model (*Niebur and Erdős, 1991*) assumes that a CPG located in the head initiates dorsoventral bends and that a combination of neuronal and sensory feedback mechanisms propagates the waves in the posteriorward direction. In this model, sensory feedback plays a modulatory role in producing smoother curvature waves but is not explicitly required for rhythm generation itself. Other computational models have aimed to describe how the motor circuit generates rhythmicity. Several neural models for the forward-moving circuit (*Karbowski et al., 2008*; *Olivares et al., 2021*) incorporating of all major neural components and connectivity have been developed. These models included a CPG in the head based on effective cross-inhibition between ventral and dorsal groups of interneurons. In contrast, *Bryden and Cohen, 2008* developed a neural model in which each segment along the body is capable of generating oscillations. In this model, a circuit of AVB interneurons and B-type motor neurons suffices to generate robust locomotory rhythms without cross-inhibition.

Other models have examined how *C. elegans* adapts its undulatory wavelength, frequency, and amplitude as a gait adaptation to external load (*Boyle et al., 2012*; *Denham et al., 2018*; *Izquierdo and Beer, 2018*; *Johnson et al., 2021*). To account for these changes, these models combined the motor circuit model with additional assumptions of stretch sensitivity in motor neurons, and worm body biomechanical constraints, to create a model that reproduced the changes in undulatory wave patterns under a range of external conditions.

Previous detailed models of *C. elegans* locomotion have employed a relatively large number of free parameters (up to 40; *Boyle et al., 2012*; *Karbowski et al., 2008*). In our work, we sought to develop a compact phenomenological model to describe an overall mechanism of rhythm generation but not the detailed dynamics of specific circuit elements. To improve predictive power, we aimed to minimize the number of free parameters used in the model. Our model has only five free parameters, yet accurately describes a wide range of experimental findings including the nonsinusoidal dynamics of free locomotion, phase response curves to transient paralysis, and dependence of frequency and amplitude on external viscosity.

Our phase portrait analysis of worm's free locomotory dynamics has described a previously undescribed methods for measuring the bending relaxation time scale $\tau_u$ and the muscle moment transition time scale $\tau_m$ (see *Appendix* for details), which may be compared with previous studies of worm biomechanics (*Fang-Yen et al., 2010*; *Berri et al., 2009*) and neurophysiology (*Milligan et al., 1997*). *Fang-Yen et al., 2010* measured out a linear relationship between the bending relaxation time scale and the external viscosity by deforming the worm body in Newtonian fluids with varied

viscosities in the range 1–25 mPa·s. Through an extrapolation based on that linear relationship, the relaxation time scale in 17% dextran NGM fluid (approximately 120 mPa·s in viscosity) is estimated to be $\approx 282ms$, which is quite close to our measured result, $\tau_u \approx 260ms$. Furthermore, our measurement of the muscle moment transition time scale ($\tau_m \approx 100ms$) is consistent with a previously measured value for muscle time scale (*Milligan et al., 1997*) that has been widely adopted for other detailed models of nematode locomotion (*Boyle et al., 2012*; *Bryden and Cohen, 2008*; *Butler et al., 2015*; *Chen et al., 2011*; *Denham et al., 2018*; *Izquierdo and Beer, 2018*; *Johnson et al., 2021*; *Karbowski et al., 2008*; *Olivares et al., 2021*; *Wen et al., 2012*).

In our model, the mechanism for generating rhythmic patterns can be characterized by a 'relaxation oscillation' process which contains two alternating sub-processes on different time scales: a long relaxation process during which the motor system varies toward an intended state due to its biomechanics under a constant active muscle moment, alternating with a rapid period during which the active muscle moment switches to an opposite state due to a proprioceptive thresholding mechanism.

The term 'relaxation oscillation', as first employed by van der Pol, describes a general form of self-sustained oscillatory system with intrinsic periodic relaxation/decay features (*Van der Pol, 1926*). The Fitzhugh-Nagumo model (*Fitzhugh, 1961*; *Nagumo et al., 1962*), a prototypical model of excitable neural systems, was originally derived by modifying the van der Pol relaxation oscillator equations. These and similar relaxation oscillators have been characterized in various dynamical systems in biology and neuroscience (*Izhikevich, 2007*). For example, the dynamics exhibited from the action potentials of barnacle muscles in their oscillatory modes were found to yield 'push-pull' relaxation oscillation characteristics (*Morris and Lecar, 1981*). The beating human heart was found to behave as a relaxation oscillator (*Der pol b, 1940*). Several studies of walking behavior in stick insects (*Bässler, 1977*; *Cruse, 1976*; *Graham, 1985*; *Wendler, 1968*) proposed that the control system for rhythmic step movements constitutes a relaxation oscillator in which the transitions between leg movements is determined by proprioceptive thresholds.

Key properties shared by these relaxation oscillators are that their oscillations greatly differ from sinusoidal oscillations and that they all consist of a certain feedback loop with a 'discharging property'. They contain a switch component that charges an integrating component until it reaches a threshold, then discharges it again (*Nave, 2007*), then repeats. Many relaxation oscillators, including the van der Pol and Rayleigh models, exhibit sawtooth-shaped phase response curves (*Der pol b, 1940*; also see *Figure 5—figure supplement 1*). As shown in our experimental and model results, all the above properties have been revealed in the dynamics of *C. elegans* locomotive behavior, consistent with the idea that the worm's rhythmic locomotion also results from a type of relaxation oscillator.

In our computational model, a proprioceptive component sensing the organism's changes in posture is required to generate adaptive locomotory rhythms. What elements in the motor system could be providing this feedback? Previous studies have suggested that head and body motor neurons, including the SMDD head motor neurons and the B-type motor neurons, have proprioceptive capabilities (*Wen et al., 2012*; *Yeon et al., 2018*) and may also be involved in locomotory rhythm generation (*Fouad et al., 2018*; *Gao et al., 2018*; *Kaplan et al., 2020*; *Xu et al., 2018*). This possibility is consistent with earlier hypothesis that the long undifferentiated processes of these cholinergic neurons may function as proprioceptive sensors (*White et al., 1986*). In particular, recent findings (*Yeon et al., 2018*) have revealed that SMDD neurons directly sense head muscle stretch and regulate muscle contractions during oscillatory head bending movements.

In our model, the proprioceptive feedback variable depends on both the curvature and the rate of change of curvature. Many mechanoreceptors are sensitive primarily to time derivatives of mechanical strain rather than strain itself; for example, the *C. elegans* touch receptor cells exhibit such a dependence (*Eastwood et al., 2015*; *O'Hagan et al., 2005*). The ability of mechanosensors to sense the rate of change in *C. elegans* curvature has been proposed in an earlier study (*Butler et al., 2015*) in which it was hypothesized that the B-type motor neurons might function as a proprioceptive component in this manner. Mechanosensors encoding a simultaneous combination of deformation and velocity have been observed in mammalian systems including rapidly-adapting (RA) and intermediate-adapting (IA) sensors in the rat dorsal root ganglia (*Rugiero et al., 2010*). Proprioceptive feedback that involves a linear combination of muscle length and velocity was also suggested by a study of *C. elegans* muscle dynamics during swimming, crawling, and intermediate

forms of locomotion (*Butler et al., 2015*). In our phenomenological model, the motor neuron constituent may represent a collection of neurons involved in motor rhythm generation. Therefore, the proprioceptive function posited by our model might also arise as a collective behavior of curvature-sensing and curvature-rate-sensing neurons.

Further identification of the neuronal substrates for proprioceptive feedback may be possible through physiological studies of neuron and muscle activity using calcium or voltage indicators. Studies of the effect of targeted lesions and genetic mutations on the phase response curves will also help elucidate roles of specific neuromuscular components within locomotor rhythm generation.

In summary, our work describes the dynamics of the *C. elegans* locomotor system as a relaxation oscillation mechanism. Our model of rhythm generation mechanism followed from a quantitative characterization of free behavior and response to external disturbance, information closely linked to the structure of the animal's motor system (*Gutkin et al., 2005*; *Nadim et al., 2012*; *Schultheiss et al., 2011*; *Smeal et al., 2010*). Our findings represent an important step toward an integrative understanding of how neural and muscle activity, sensory feedback control, and biomechanical constraints generate locomotion.

# Materials and methods

## Key resources table

| Reagent type (species) or resource | Designation | Source or reference | Identifiers | Additional information |
|---|---|---|---|---|
| Strain, strain background (*E. coli*) | OP50 | CGC | Fang-Yen Lab Strain Collection: OP50 RRID:WB-STRAIN:WBStrain00041971 | OP50 |
| Strain, strain background (*C. elegans*) | YX148 | *Fouad et al., 2018* | Fang-Yen Lab Strain Collection: YX148 | *qhIs1[Pmyo-3::NpHR::eCFP; lin-15(+)]; qhIs4[Pacr-2::wCherry]* |
| Strain, strain background (*C. elegans*) | YX119 | *Fouad et al., 2018* | Fang-Yen Lab Strain Collection: YX119 | *qhIs1[Pmyo-3::NpHR::eCFP; lin-15(+)]; unc-49(e407)* |
| Strain, strain background (*C. elegans*) | YX205 | *Leifer et al., 2011* | Fang-Yen Lab Strain Collection: YX205 | *hpIs178[Punc-17::NpHR::eCFP; lin-15(+)]* |
| Strain, strain background (*C. elegans*) | WEN001 | *Fouad et al., 2018* | Fang-Yen Lab Strain Collection: WEN001 | *wenIs001[Pacr-5::Arch::mCherry; lin-15(+)]* |

## Worm strains and cultivation

*C. elegans* were cultivated on NGM plates with *Escherichia coli* strain OP50 at 20°C using standard methods (*Sulston and Hodgkin, 1988*). Strains used and the procedures for optogenetic experiments are described in the Key resources table and *Appendix*. Preparation of OP50 and OP50-ATR plates were as previously described (*Fouad et al., 2018*). All experiments were performed with young adult (< 1 day) hermaphrodites synchronized by hypochlorite bleaching.

## Locomotion and phase response analyses

To perform quantitative recordings of worm behavior, we used a custom-built optogenetic targeting system as previously described (*Fouad et al., 2018*; *Leifer et al., 2011*). Analysis of images for worm's body posture was performed using a previously developed custom software (*Fouad et al., 2018*). The anterior curvature is defined as the average of the curvature over body coordinate 0.1–0.3; excluding the range from 0 to 0.1 avoided measurement of high-frequency movements of the worm's anterior tip. Descriptions of the apparatus and image analyses are available in *Appendix*.

For phase response experiments, opsin-expressing worms were illuminated using a brief laser pulse (532 nm wavelength, 0.1 or 0.055 s duration, irradiance 16 mW/mm$^2$) in the head region (0–0.25 body coordinate). A total of 10 trials with 6 s interval between successive pulses were performed for each animal. Trials in which the worms did not maintain forward locomotion were censored. To generate the phase response curve (PRC), we calculated the phase of inhibition of each trial and the resulting phase shift. Details of calculations for the averaged PRC are given in *Appendix*.

All the data and image analysis codes used in the manuscript are available at Dryad (archived at https://doi.org/10.5061/dryad.wwpzgmsk2).

## Computational modeling

Our primary model is based on a novel neural control mechanism incorporated with a previously described biomechanical framework (*Fang-Yen et al., 2010*; *Gray and Lissmann, 1964*; *Wallace, 1968*). A proprioceptive signal is defined by a linear combination of bending curvature and rate of change of curvature. When the signal reaches a threshold, a switching command is initiated to reverse the direction of muscle moment. The worm's curvature relaxes toward the opposite direction, and the process repeats, creating a dorsoventral alternation. Detailed descriptions of implementation and fitting procedure of this model and alternative models are available in *Appendix*. All codes for modeling analyses are available at Dryad (https://doi.org/10.5061/dryad.wwpzgmsk2).

## Acknowledgements

We thank Mei Zhen and Quan Wen for providing strains. Some strains were provided by the CGC, funded by NIH Office of Research Infrastructure Programs (P40 OD010440). We thank Gal Haspel, Michael Carchidi, and Patrick McClanahan for helpful discussions. HJ, ADF, and CF-Y. were supported by the National Institutes of Health (1R01NS084835). ST was supported by an Abraham Noordergraaf Research Fellowship and a Littlejohn Fellowship.

## Additional information

### Funding

| Funder | Grant reference number | Author |
|--------|------------------------|--------|
| National Institutes of Health | 1R01NS084835 | Hongfei Ji<br>Anthony D Fouad<br>Christopher Fang-Yen |

The funders had no role in study design, data collection and interpretation, or the decision to submit the work for publication.

### Author contributions

Hongfei Ji, Data curation, Software, Formal analysis, Investigation, Visualization, Methodology, Writing - original draft, Writing - review and editing; Anthony D Fouad, Software, Investigation, Methodology, Writing - original draft, Writing - review and editing; Shelly Teng, Alice Liu, Pilar Alvarez-Illera, Zihao Li, Investigation; Bowen Yao, Software, Investigation; Christopher Fang-Yen, Conceptualization, Resources, Software, Supervision, Funding acquisition, Methodology, Writing - original draft, Project administration, Writing - review and editing

### Author ORCIDs

Hongfei Ji https://orcid.org/0000-0001-9617-6411
Anthony D Fouad http://orcid.org/0000-0002-4677-2968
Zihao Li https://orcid.org/0000-0003-4304-8322
Christopher Fang-Yen https://orcid.org/0000-0002-4568-3218

### Decision letter and Author response

Decision letter https://doi.org/10.7554/eLife.69905.sa1
Author response https://doi.org/10.7554/eLife.69905.sa2

## Additional files

### Supplementary files

• Transparent reporting form

**Data availability**

All data and software have been deposited to a Dryad repository.

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

## Appendix 1

### Strains and plates preparations

Transgenic strains used in this study are listed as follows: YX148 *qhIs1[Pmyo-3::NpHR::eCFP;lin-15(+)];qhIs4[Pacr-2::wCherry]*, YX119 *unc-49(e407);qhIs1[Pmyo-3::NpHR::eCFP;lin-15(+)]*, YX205 *hpIs178[Punc-17::NpHR::eCFP;lin-15(+)]*, WEN001 *wenIs001[Pacr-5::Arch::mCherry;lin-15(+)]*.

For optogenetic experiments, worms were cultivated in darkness on plates with OP50 containing the cofactor all-*trans* retinal (ATR). For control experiments and free-moving experiments, worms were cultivated on regular OP50 NGM plates without ATR. To make OP50-ATR plates, we added 2 µL of a 100 mM solution of ATR in ethanol to an overnight culture of 250 µL OP50 in LB medium and used this mixture to seed 6 cm NGM plates.

### Locomotion analysis

To analyze worm locomotion in viscous fluids, we placed worms in dextran solutions in chambers formed by a glass slide and a coverslip separated by 125-µm-thick polyester shims (McMaster-Carr 9513K42). For viscosity-dependence experiments, we used 5%, 17%, and 35% (by mass) solutions of dextran (Sigma-Aldrich D5376, average molecular weight 1,500,000–2,800,000) in NGMB. These solutions were measured to have viscosities of 10, 120, and 5400 mPa·s (*Fang-Yen et al., 2010*), respectively. We used a 17% dextran solution for all other experiments. NGMB consists of the same components as NGM media (*Stiernagle, 2006*), but without agar, peptone, or cholesterol.

We recorded image sequences using a custom-built optogenetic targeting system based on a Leica DMI4000B microscope under 10X magnification with dark field illumination provided by red LEDs. Worm images were recorded at 40 Hz with an sCMOS camera (Photometrics optiMOS). We used a custom-written C++ software (*Fouad et al., 2018*) to perform real-time segmentation of the worm during image acquisition. The worm was identified in each image by its boundary and center-line calculated from a binary image. Anterior-posterior orientation was noted visually during the recording. Segmentation information, including coordinates of the worm boundary and centerline, was saved to disk along with the corresponding image sequences.

Post-acquisition image analysis was performed using a custom MATLAB (Mathworks) similar to previous reports (*Fouad et al., 2018*). The worm centerline of each image was smoothed using a cubic spline fit. We calculated curvature $\kappa$ as the dot product between the unit normal vector to the centerline and derivative of the unit tangent vector to the centerline with respect to the body coordinate. Dimensionless curvature $K$ was calculated as the product of $\kappa$ and the worm body length $L$ represented by length of the centerline. Since the segmentation was relatively noisy at the tips of the worm, we excluded curvature in the anterior and posterior 5% of the body length. The worm's direction of motion was identified by calculating the gradients in the curvature over time and body coordinate, and image sequences in which the worm performed consistent forward movement (lasting at least 4 s) were selected for analysis. The anterior curvature $K(t)$ was defined as the average of the dimensionless curvature over body coordinate 0.1-0.3; this range avoided high-frequency movements of the anterior tip of the animal.

To quantify oscillatory dynamics during forward locomotion, we identified undulatory cycles from the time sequence of anterior curvature in each worm. Local extrema along each sequence were identified and portions between consecutive local maxima were defined as individual cycles. To minimize the effects of changes in the worm's frequency, we excluded cycles whose period deviated by more than 20% from the average period of all worms' undulations in each experimental session.

For the ease of computing average dynamics, we converted individual cycles from a time-dependent to a phase-dependent curvature by uniformly rescaling each cycle to a phase range of $2\pi$. The averaged curvature within one cycle was then computed by averaging all individual cycles in the phase domain: $\langle K(\phi) \rangle = \sum_{i=1}^{N} K_i(\phi)/N$. Similarly, the averaged phase derivative of curvature within one cycle was calculated as $\langle dK/d\phi \rangle = \sum_{i=1}^{N} (dK_i/d\phi)/N$.

### Stability of the worm's head oscillation

To examine the stability of the worm's head oscillation during forward locomotion, we analyzed head oscillations of worms that were optogenetically perturbed with 0.1 s muscle inhibitions and

estimated their recovery dynamics after being deviated from the normal oscillation due to the perturbation.

To illustrate the oscillation dynamics, we use a two-dimensional variable, $\boldsymbol{x} = (K, \xi\dot{K})$ in the unit of curvature where $\xi = 0.135s$ is a scaling factor. In *Figure 3—figure supplement 2*, we depicted the closed trajectory (black) in the plane spanned by the variables $K$ and $\xi\dot{K}$ for the head oscillation of unperturbed moving worms (this coordinate plane is in fact a linearly scaled version of the phase plane spanned by the variables $K$ and $\dot{K}$), which we call as the normal cycle of the worm's head oscillation.

Next, we defined an amplitude variable $d$ that represents the normalized deviation to the normal cycle. If the oscillator is stable, the closed orbit for the unperturbed dynamics is usually called the stable limit cycle. Here, we stick to the notion of normal cycle instead of using 'limit cycle' to avoid any confusion on the stability of the worm's head oscillation. For any phase state of an individual oscillation, the normalized deviation to the normal cycle is defined as $d(\phi) = (D(\phi) - D^C(\phi))/D^C(\phi)$. Here, $D(\phi)$ is distance to the center of oscillation on the phase plane, which is set to the origin, such that $D(\phi) = \sqrt{K(\phi)^2 + (\xi\dot{K}(\phi))^2}$ where $\phi$ denotes the phase value of the current state estimated by the four-quadrant inverse tangent of the variable pair $(K, \xi\dot{K})$. In this expression, $D^C(\phi)$ denotes the distance to the center of oscillation that is evaluated exactly on the normal cycle at phase $\phi$.

Using the deviation to the normal cycle to describe the amplitude of the worm's head oscillation, we collected the amplitude dynamics over time for all periods of the worm's head oscillations during which no illumination pulse occurs, that is, all periods of locomotion between two consecutive illumination pulses. We grouped the amplitude dynamics into bins according to their initial amplitudes and then calculated the collective amplitude dynamics for each bin. As shown in *Figure 3—figure supplement 1*, the collective amplitude variable $d$ converges to zero after roughly 0.5 s regardless of the initial amplitude. This result indicates that the worm's head oscillation returns to its normal oscillation after being perturbed and that the normal cycle may represent a stable limit cycle for the oscillation.

## Phase isochron map and vector field for the worm's head oscillation

On the normal cycle we define the phase of the oscillation as $\phi^C(t) = \omega_0 \cdot t_{modT_0}$, where $\omega_0 = 2\pi/T_0$ is the angular frequency of normal oscillation (the calculation of $T_0$ will be described in the next subsection) and we determined the initial phase ($\phi^C = 0$) to be when $K$ reaches local maximum (or $\boldsymbol{x} = (K_{max}, 0)$) and hence $\phi^C = \pi$ to be when $K$ reaches local minimum (or $\boldsymbol{x} = (K_{min}, 0)$). In this way, we parameterized the normal cycle by defining a bijective map between phases and state points $\Phi(\boldsymbol{x}^C) = \phi^C$.

The map $\Phi(\boldsymbol{x}) = \phi$ has been well defined for all the state points on the normal cycle $C$. We next estimate the phases for points off the normal cycle. By definition (*Izhikevich, 2007*), if $\boldsymbol{x}_0$ is a point on the normal cycle and $\boldsymbol{y}_0$ is a point off the normal cycle, then $\boldsymbol{y}_0$ will have the same phase as $\boldsymbol{x}_0$ if the trajectory starting at the initial point $\boldsymbol{y}_0$ off the normal cycle converges to the trajectory starting at the initial point $\boldsymbol{x}_0$ on the normal cycle as time goes to infinity. Here, we define the set of all state points off the normal cycle having the same phase as a point $\boldsymbol{x}_0$ on the normal cycle as the isochron (*Winfree, 2001*) for phase $\phi_0 = \Phi(\boldsymbol{x}_0)$.

In our analysis, it was not possible to define an isochron according to the theoretical definition since data were always recorded in a finite time period during experiments. We used an alternative way to estimate all isochrons on the phase plane for the worm's head oscillation. For each individual trial of illumination, we observed that, due to the optogenetic inhibition, the variable $\dot{K}$ quickly decayed toward zero value immediately after the illumination and then recovered after approximately 0.3 s as the oscillation converged to a normal oscillation. Therefore, by finding the local minimal of $\dot{K}$ immediately after each illumination pulse, we located the point at which the paralyzing effect is just removed and after which the oscillation starts a free resumption to normal oscillation. We call this point the 'notch point' $\boldsymbol{x}_N$ as it can be clearly seen from the phase plot (as shown in *Figure 3E, H and K*). After the notch point $\boldsymbol{x}_N$, the oscillation will proceed to its next phase states $\boldsymbol{x}(\phi = 0)$ and $\boldsymbol{x}(\phi = \pi)$ (or vice-versa), both of which can be easily identified through peak finding from the curvature dynamics $K$. Hence, we obtained two sub-trajectories from the oscillation: one

$x_N \to x(\phi = 0)$, and the other $x_N \to x(\phi = \pi)$. Next to determining the timing of the notch point $t(x_N)$, we determined the phase of the notch point in the following steps: (1) we computed the phase of the state on the normal cycle at time $t(x_N)$ as if the perturbation had not occurred, which is $\phi_u(x_N^C) = \omega_0(t(x_N) - t(x(\phi = 0)))_{modT_0}$, or $\phi_l(x_N^C) = \omega_0(t(x_N) - t(x(\phi = \pi)))_{modT_0} - \pi$. Here, $\phi(x_N^C)$ was computed twice using phase states $x(\phi = 0)$ [subscripted with $u$] and $x(\phi = \pi)$ [subscripted with $l$] as references, respectively; (2) we calculated the induced phase shift $PRC(t_{illum})$ and the phase of the notch point is $\phi(x_N) = \phi(x_N^C) - PRC(t_{illum})$. After obtaining the sub-trajectories $x_N \to x(\phi = 0)$ and $x_N \to x(\phi = \pi)$ and calculating the phase of $x_N$, we then estimated the phase values for all the points within each of the two sub-trajectories through linear interpolation.

Following the above steps, we calculated the phase values for all the state points on the phase plane that have been recorded from the optogenetic experiments. We then applied a 2-D moving average (using the angular statistics method) for the obtained phase values over the phase plane to smooth out the isochron map. Finally, we used a linear 2-D interpolation to obtain a phase isochron map with a finer resolution as shown in *Figure 3—figure supplement 2*.

To compute the vector field of the worm's head oscillation, we collected all the sub-trajectories $x_N \to x(\phi = 0)$ and $x_N \to x(\phi = \pi)$ that are defined above and took derivative of the trajectories with respect to time. Thus, by collecting all the phase states $(K, \xi\dot{K})$ and their corresponding time derivatives $(dK/dt, d(\xi\dot{K})/dt)$ that describe the tangent vectors of trajectories, we generated the raw form of vector field for the worm's head oscillation. Again, we applied a 2-D moving average for the raw outcome over the phase plane to smooth out the vector field. We used a linear 2-D interpolation to obtain a vector field with an appropriate number of quivers to be displayed (*Figure 3—figure supplement 2*).

## Phase response analysis

To generate phase response curves (PRCs) from optogenetic inhibition experiments, each trial's illumination phase $\phi$, as well as the induced phase shift $F$, were calculated. To calculate the two variables, the animal's phase of oscillation was estimated based on timings of local extrema identified from the time-varying curvature profiles via a peak finding method. Specifically, (i) the occurrence of illumination of the trial was set to $t = T_{illum}$; $t = 0$ was set at the beginning of each experiment. (ii) Around the illumination, timings of the two local maxima of curvature immediately before and after were identified as the two zero-phase points of the oscillation before and after the illumination, respectively. Here, the timings are denoted as $TZ_{-2}$, $TZ_{-1}$, $TZ_{+1}$, and $TZ_{+2}$, in the ascending order of time. (iii) Similarly, timings of the two local minima of curvature immediately before and after the illumination were identified as the two half-cycle-phase points before and after the illumination, respectively. Here, the timings are denoted as $TH_{-2}$, $TH_{-1}$, $TH_{+1}$, and $TH_{+2}$, in the ascending order of time. (iv) With these measurements, cycle period $T_0$ was computed as $T_0 = (TZ_{+2} - TZ_{+1} + TZ_{-1} - TZ_{-2} + TH_{+2} - TH_{+1} + TH_{-1} - TH_{-2})/4$, so the angular frequency of undulation $\omega_0 = 2\pi/T_0$ ($T_0$ was computed as the average of differences of adjacent local maxima/minima before and after illumination; multiple cycles were used here to reduce noise). In addition, the illumination phase $\phi$ of each individual trial was computed as $\phi_u = \omega_0(T_{illum} - TZ_{-1})_{modT_0}$, $\phi_l = \omega_0(T_{illum} - TH_{-1} + T_0/2)_{modT_0}$, and the corresponding phase shift $F$ was computed as $F_u = \omega_0(TZ_{+1} - TZ_{-1})_{modT_0} - \pi$, $F_l = \omega_0(TH_{+1} - TH_{-1} + T_0/2)_{modT_0} - \pi$. Here, phase of illumination and the corresponding phase shift were computed twice using zero [subscripted with $u$] and half-cycle [subscripted with $l$] phase points as references, respectively.

We generated 2-D scatter plots for all trials with illumination phase as $x$ coordinate and the corresponding phase shift as $y$ coordinate. To visualize the distribution of the scatter points we generated bivariate histogram plots by grouping the data points into 2-D bins with 25 bins on both dimensions covering the range $[0, 2\pi]$ for $x$ and range $[-\pi, \pi]$ for $y$. To indicate average tendency of phase shift depending on phase of illumination, we calculated a mean-curve representation of PRCs via a moving average operation. In this process, each mean was calculated over a sliding window of width $0.16\pi$ along the direction of $\phi$ from 0 to $2\pi$. The 95% confidence interval relative to each window of data points was also computed and an integral number of them were displayed as filled area around the PRC. Through the computation, all statistical calculations followed the rules of directional statistics (*Fisher et al., 1993*) since $\phi$ and $F$ are circular variables defined in radians.

## Phase response curves from perturbations of other body regions

We asked how phase responses for the other regions of the body would compare to that of the anterior region. We conducted optogenetic experiments that inhibited *Pmyo-3::NpHR* transgenic worms by transiently illuminating 0.1–0.3 (anterior), 0.4–0.6 (middle), and 0.6–0.8 (posterior) of the body length, respectively. We found that the amplitude of the sawtooth feature of PRC tends to decrease as the perturbation occurs further from the head (*Figure 3—figure supplement 5A,E,I*). We also noticed that, for the same perturbed region, the PRC shape remains unaffected regardless of the region at which the dynamics were analyzed (see *Figure 3—figure supplement 5A–C,D–F,G–I*, respectively), suggesting that posterior regions of a freely moving worm follow their anterior neighbors with a constant phase offset. Taken together, these results suggest that a main rhythm generator may operate near the head of the worm to produce primary oscillations during forward locomotion. The sawtooth-shape feature of the PRC becomes stronger if the perturbation hits closer to the rhythm generator (*Figure 3—figure supplement 5A*) or becomes weaker if the perturbation indirectly affects it (*Figure 3—figure supplement 5E,I*)

## The relaxation oscillator model for locomotor wave generation

We first developed a relaxation oscillator model to simulate the rhythm generation during *C. elegans* forward locomotion. In this model, we incorporated a novel neuromuscular mechanism with a previously described biomechanical framework (*Fang-Yen et al., 2010*). Here, we only simulated the bending rhythms generated from the head region; the wave propagation dynamic is out of the scope of our study. Our phenomenological model does not contain detailed activities of individual neurons but focus on describing key neuromuscular mechanisms and their contributions to the rhythm generation.

To produce model variables that can be directly compared with experimental observations of moving animals, a biomechanical framework was first developed to describe worm's behavioral dynamics in its external environments. Following previous derivations for *C. elegans* biomechanics (*Fang-Yen et al., 2010*), the relationship between animal behavioral outputs and the active muscle moments can be described as follows:

$$C_N \frac{\partial y}{\partial t} + a \frac{\partial^2 \kappa}{\partial s^2} + a_v \frac{\partial}{\partial t}\left(\frac{\partial^2 \kappa}{\partial s^2}\right) = m_a. \tag{A1}$$

In *Equation A1*, the first term indicates the external viscous force that is transverse to the body segment where $C_N$ is the coefficient of viscous drag to the transverse movement and $y$ denotes the lateral displacement of body segment; the second term indicates the internal elastic force where $a$ is the bending modulus of the worm body; the third term indicates the internal viscous force where $a_v$ is the coefficient of the body internal viscosity. On the right side of *Equation A1* is the active muscle moment $m_a$.

Taking the second partial derivative with respect to body coordinate $s$ on both sides of *Equation A1* and, using the linear relation under the small-amplitude approximation, $\kappa \approx y_{ss}$, we arrive at:

$$C_N \frac{\partial \kappa}{\partial t} + a \frac{\partial^4 \kappa}{\partial s^4} + a_v \frac{\partial}{\partial t}\left(\frac{\partial^4 \kappa}{\partial s^4}\right) = \frac{\partial^2 m_a}{\partial s^2}. \tag{A2}$$

Under the assumptions of small-amplitude undulations and a fixed wavelength $\lambda$ down the worm body, $\kappa$ can be considered as a travelling sinusoidal wave with a small deviation, $\kappa(s,t) = \kappa_0 \sin(2\pi s/\lambda - \omega t) + \delta$, which leads to an approximation, $\kappa_{ssss} \approx (2\pi/\lambda)^4 \kappa$. Plugging this approximation into *Equation A2* while keeping $s$ fixed, after some rearrangement, one gets:

$$\kappa + \frac{C_N \left(\frac{\lambda}{2\pi}\right)^4 + a_v}{a} \dot{\kappa} = \frac{\lambda^4}{(2\pi)^4 a} \frac{\partial^2 m_a}{\partial s^2}. \tag{A3}$$

In terms of the dimensionless curvature $K = \kappa \cdot L$ and dimensionless muscle moment

$$M_a = \frac{\lambda^4 L}{(2\pi)^4 a}\frac{\partial^2 m_a}{\partial s^2}, \tag{A4}$$

we can rewrite *Equation A3* as:

$$K + \tau_u \dot{K} = M_a, \tag{A5}$$

where

$$\tau_u = \frac{C_N\left(\frac{\lambda}{2\pi}\right)^4 + a_v}{a}, \tag{A6}$$

and we note that *Equations A5 and A6* yield *Equation 1*. In *Equation A6*, both the wavelength $\lambda$ and the normal viscous drag coefficient $C_N$ vary with the fluid viscosity $\eta$ (*Berri et al., 2009* ; *Fang-Yen et al., 2010*).

The above biomechanical framework in our model treats the worm's body segment as a viscoelastic rod and describes how the body segment bends under the forces provided by the active muscle moment. However, the simulated oscillation in $K$ comes from the rhythmicity of the active muscle moment which originates from the hypothesized neuromuscular mechanism described by the following relaxation-oscillation process:

i.   Proprioceptive feedback is sensed as a linear combination of the current curvature value and the current rate of change of curvature, $P = K + b\dot{K}$ (black curve in *Figure 4D*).
ii.  During movement of bending, this proprioceptive feedback is constantly compared with two threshold values $P_{th}$ and $-P_{th}$ (gray dashed bars in *Figure 4D*).
iii. Once the feedback reaches either of the thresholds (the switch points as indicated by red circles in *Figure 4D*), a switch command is initiated (blue square wave in *Figure 4E*).
iv.  The switch command triggers the active muscle moment to change toward the opposite saturation value (black curve in *Figure 4E*).

To simulate the switch-triggered muscle transition we used a modified logistic function: $M_a(t) = \pm M_0 \cdot \tanh(t/2\tau_m)$. Here, the plus sign indicates the dorsal-to-ventral muscle moment transition while the minus sign indicates the opposite direction.

To initiate the oscillation in our model we set the system to bend toward the ventral side by setting $M_a|_{t=0} = M_0$ and $K|_{t=0} = 0$. During forward locomotion, the active muscle moment oscillates by undergoing a relaxation oscillation process: a relaxation subperiod during which $M_a$ stays at a saturated bending state ($M_0$ for ventral bending, $-M_0$ for dorsal bending), alternating between a shorter subperiod during which $M_a$ quickly transits toward the opposite state due to effects described in iii and iv. The bending curvature $K(t)$ which is driven by $M_a$ in an exponential decaying manner (*Equation A5*) follows the rhythmic activity of $M_a$, thereby also exhibiting an oscillatory dynamic (*Figure 4B*).

This relaxation oscillator model reproduces two key features of free locomotion that we observed from experiments. First, freely moving worms exhibit nonsinusoidal curvature waveform with an intrinsic asymmetry: bending toward the ventral or dorsal directions occurs slower than straightening toward a straight posture during each locomotory cycle (*Figure 4F*). Second, dynamic of the active muscle moment shows a trapezoidal waveform during forward locomotion (*Figure 2D Inset* and *Figure 4E*). These results are independent of external conditions but reflect intrinsic properties of the neuromuscular mechanisms underlying locomotion rhythm generation.

Note that parameters $M_0$, $\tau_u$, and $\tau_m$ were estimated from data of free locomotion using phase portrait techniques described in the following subsection. Parameters $b$ and $P_{th}$ were yet degenerate in this model of free locomotion. Here, we temporarily set $b = 0$ and then set $P_{th}$ such that the oscillatory period predicted by model matched the average period measured from experiments with a minimum squared error:

$$P_{th} = \underset{P_{th}>0}{\operatorname{argmin}}\left(T_{model}(P_{th}) - T_{experiment}\right)^2. \tag{A7}$$

The nondegeneracy of $b$ and $P_{th}$ was determined by fitting the model to the experimental PRC as described in the later subsection so that all the parameters for the model are provided as $M_0 = 8.45$, $\tau_u = 260\,ms$, $\tau_m = 100\,ms$, $b = 46\,ms$, and $P_{th} = 2.33$.

## Measuring bending relaxation time scale and amplitude of active muscle moment

To estimate these two parameters, we applied a heuristic method that uses the shape properties of *C. elegans* free-running phase plot *(Figure 2D)*. From the curve in the figure, we noticed two 'flat' portions symmetrically distributed at quadrant *I* and *III* on the phase plane. Recalling *Equation 1* (or *Equation A5*): $K + \tau_u \dot{K} = M_a(t)$, the two flat regions indicate that the scaled active muscle moment, $M_a(t)$, is nearly constant during the corresponding time bouts.

We then computed the linear correlation between variables $K$ and $\dot{K}$ to identify the two 'flat' regions and, through linear fits, obtained two linear relations respectively: $\langle K \rangle + \tau_1 \langle \dot{K} \rangle = M_1$ (region 1) and $\langle K \rangle + \tau_2 \langle \dot{K} \rangle = M_2$ (region 2). Thus, the bending relaxation time scale $\tau_u$ and the amplitude of the scaled active muscle moment are estimated as $\hat{\tau}_u = (\tau_1 + \tau_2)/2$ and $\hat{M}_0 = (|M_1| + |M_2|)/2$, respectively.

The above method used the phase plot measured from locomotion of worms swimming in a 17% dextran solution (120 mPa·s viscosity) as an example. However, it is also valid for estimating parameters of locomotion in other viscosities.

## Measuring active moment transition time scale

With $\tau_u$ (estimated from the above method), $\langle K \rangle$ and $\langle \dot{K} \rangle$ (measured from locomotion) plugged to the left side of *Equation 1*, we were able to compute the waveform of the scaled active muscle moment $M_a(t)$ on the right side of *Equation 1*. As expected and shown in *Figure 2D Inset*, the curve of $M_a(t)$ is roughly centrally symmetric around point $(T_0/2, 0)$ on the plane, with two plateau portions indicating two saturated states for dorsal and ventral muscle contractions, respectively.

Between the two plateau portions represents a period during which the active muscle moment is undergoing a ventral-to-dorsal (or vice-versa) transition. We used a modified logistic function to model the ventral-to-dorsal muscle moment transition (substituting $t$ with $-t$ for transition in the other direction):

$$M_a(t) = M_0 \cdot \tanh\left(\frac{t}{\tau_m}\right). \tag{A8}$$

To estimate $\tau_m$, the exponential time constant for the transition of active muscle moment, we took the time derivative of *Equation A8* and took the absolute value of the resultant:

$$\left|\frac{dM_a}{dt}\right| = \frac{M_0}{\tau_m} \cdot \frac{\exp(2t/\tau_m)}{(1+\exp(2t/\tau_m))^2}. \tag{A9}$$

We notice that when $t = 0$, the maximum of $|dM_a/dt|$ is achieved and the value is $M_0/\tau_m$. On the other hand, the maximum of $|dM_a/dt|$ can be obtained from the experimental observation by simply finding the peak of $|dM_a/dt|$ curve where $M_a = \langle K(t) \rangle + \tau_u \cdot \langle dK(t)/dt \rangle$. Thus, $\tau_m$ can be estimated as:

$$\tau_m = \hat{M}_0 \cdot \left|\frac{dM_a}{dt}\right|^{-1}_{max}. \tag{A10}$$

## Parameter estimation

For our original threshold-switch model, parameters $\tau_u$, $\tau_m$, and $M_0$ were estimated from free locomotion experiments as described above. These three parameters nearly fully determine the biomechanical framework of *C. elegans* bending movements (governed by *Equation A5 and A8*). On the other hand, parameters $b$ and $P_{th}$ describe the proprioceptive feedback and the threshold-switch features in our model. Specifically, they characterize two threshold lines, $K + b\dot{K} = \pm P_{th}$ (as shown in *Figure 4C*). The two switch points—defined by the intersection between the phase trajectory and the threshold lines on the phase plane—determine the timing of switches for the active muscle moment (see *Figure 4C–E*). We noted that the model behavioral output of free locomotion is degenerate with respect to these two parameters; the same outcome would be produced if the threshold lines cross the same pair of switch points. To first determine the free-moving dynamic as

well as the switch points, we temporarily set $b = 0$ and then set $P_{th}$ such that the oscillatory period defined by model matched the average period measured from the experiments.

To obtain the nondegeneracy of $P_{th}$ and $b$, we fit our model to the experimental phase response curve using a global optimization procedure. Full procedure for the determination of $b$ and $P_{th}$ is given below.

## Modeling worm oscillations in varied environments

Differences in various environments will change only those parameters that are related to contact with external forces whereas parameters related to oscillator's internal properties will not be affected. In terms of the internal parameters of our model, we used values that were previously determined, which are $\tau_m = 100\,ms$, $M_0 = 8.45$, $b = 46\,ms$, $P_{th} = 2.33$. For the exogenous parameters, only the time constant of undulation, $\tau_u$, varies according to external conditions. According to *Equation A6*, $\tau_u$ is explicitly determined in terms of other physical parameters, including bio-mechanical parameters measured in previous work (*Fang-Yen et al., 2010*): the internal viscosity of worm body is measured as $a_v = 5 \cdot 10^{-16}\,Nm^3s$; the bending modulus of worm body is measured as $a = 9.5 \cdot 10^{-14}\,Nm^3$; $C_N = 31\eta$ is the coefficient of viscous drag for movement normal to the body (*Katz et al., 1975*), where $\eta$ is the fluid viscosity. According to previous measurements of undulatory wavelengths in different viscous solutions (*Fang-Yen et al., 2010*), we applied a logarithmic fit to the data points, yielding $\lambda/L = -0.158 \log_{10}(\eta/\eta_0) + 1.5$ for a continuous model realization in undulatory frequency and amplitude. Here, $\lambda$ is the wavelength and $\eta_0 = 1\,mPa \cdot s$.

## Alternative models for locomotor wave generation

To further evaluate the performance of our original model, we explored three alternative models for simulating locomotory rhythm generation to make comparisons across these models and the experimental observations. Alternative models are based on three previously studied self-oscillator models described by 2-D nonlinear systems: the van der Pol, Rayleigh, and Stuart-Landau oscillators.

First, we developed a model oscillator in the form taken from the van der Pol Oscillator:

$$\ddot{K} + a_V(b_V K^2 - 1)\dot{K} + \omega_V^2 K = 0, \tag{A11}$$

where $K$ indicates the nondimensional bending curvature. This model has a nonlinear damping term with a coefficient depending on $K$. Simulated oscillation is a limit cycle of the model (*Figure 5—figure supplement 1B,F*), given parameters $a_V = -0.026\,s^{-1}, b_V = -2.04, \omega_V = 5.51\,s^{-1}$.

Second, we developed a model oscillator by modifying the Rayleigh Oscillator:

$$\ddot{K} + a_R(b_R \dot{K}^2 - 1)\dot{K} + \omega_R^2 K = 0, \tag{A12}$$

where $K$ again indicates the nondimensional bending curvature. This model has a nonlinear damping term with a coefficient depending on $\dot{K}$. Simulated oscillation is a limit cycle of the model (*Figure 5—figure supplement 1C,G*), given parameters $a_R = 2.73\,s^{-1}, b_R = 0.0023\,s^2, \omega_R = 5.6\,s^{-1}$.

Third, we developed a model oscillator by modifying the Stuart-Landau Oscillator:

$$\dot{Z} + \left(\frac{l}{2}|Z|^2 - \sigma\right)Z = 0. \tag{A13}$$

Here, the system is described in the complex domain where $Z = Z_r + iZ_i$, $l = l_r + il_i$ are complex variables, and $\sigma$ is real. We let $Z_r$, the real part of $Z$, denote the nondimensional curvature $K$. This model has a nonlinear damping term with coefficient depending on $|Z|$. Simulated oscillation is a limit cycle of the model (*Figure 5—figure supplement 1D,H*), given parameters $l_r = 0.54\,s^{-1}, l_i = 0.52\,s^{-1}, \sigma = 5.54\,s^{-1}$.

The three alternative models produce free-running oscillatory dynamics with similar frequency and amplitude as measured from worms swimming in fluids with viscosity $120\,mPa \cdot s$.

## Simulation of optogenetic inhibition

According to our experimental observations on the effect of the optogenetic muscle inhibition (*Figure 3A,B*), paralysis of muscles of the illuminated region initiated upon illumination (defined as

$t = 0$ for **Figure 3B**) and reached maximal effect approximately at $t = 0.3\,s$. Here, we modeled the process of muscle inhibition by multiplying the scaled active muscle moment, $M_a$, with a factor, $1 - Q(\Delta t)$, as a function of the time interval $\Delta t$ in a bell-shaped form (**Figure 4—figure supplement 1**, **Equation A14**).

As described in our model, the dorsoventrally alternating feature of the active muscle moment during locomotion are described by the dynamics of $M_a(t)$. Specifically, $M_a(t)$ is positive when ventral muscles contract and dorsal muscles relax, and negative for the other half of the cycle. Therefore, in our threshold-switch model, specifically inhibiting dorsal- or ventral- or both-side muscles was computationally equivalent to conditionally modulating $M_a(t)$ with the bell-shaped modulating function depending on the sign of $M_a(t)$.

For simulating inhibition process in the three alternative models, we factored out a specific term from individual model equations as a generalized active muscle moment. We applied the bell-shaped modulating function to this term conditionally for each individual model. Detailed descriptions of implementing modeled inhibitions in alternative models are available from below.

To get a deeper understanding of how phase response curves are related to systems dynamics during wave generation, we systematically simulated transient muscle inhibitions on individual model oscillators at different times within a cycle period to generate model PRCs. To do that, we theoretically simulated the process of muscle inhibition by multiplying model active muscle moment with a modulatory factor, $1 - Q(\Delta t)$, which has a bell-shaped profile (**Figure 4—figure supplement 1**):

$$Q(\Delta t) = \frac{H}{\left(1 + \left|\frac{\Delta t - r}{p}\right|^{2q}\right)},$$
(A14)

where $r = 0.3\,s$ is the timing of the occurrence of maximal paralysis according to our experimental observations on the effect of muscle inhibition (**Figure 3A,B**), $H$ indicated the maximal degree of paralysis, and $p$, $q$ measure the paralyzing rate and duration, respectively. To ensure sufficient smoothness during computation, we let $p = 0.3 \cdot 10^{-1/q}$ so that $Q|_{\Delta t=0} > 0.99$. Note that when modeling the dorsal-side-only muscle inhibition, the parameter $H$ for describing max degree of optogenetic muscle inhibition was modulated to $H = 0.5 * H_{optimal}$ to qualitatively agree with experimental observations (**Figure 6**). This factor accounts for unequal degrees of paralysis during ventral vs. dorsal illumination (**Figure 6—figure supplement 1**), causing the PRC of dorsal-side illumination to show a relatively moderate response compared to ventral-side illumination.

To simulate the muscle inhibition on our threshold-switch model, we multiplied $M_a$ with $(1 - Q)$ any time the model was to be inhibited during its oscillatory period. To apply this operation to the alternative models, we factored out a term as a generalized active muscle moment for each individual model and then multiplied it with the bell-shaped function described above. The generalized forms of active muscle moment for the alternative models are implemented by modifying their original forms as follows:

a. For the van der Pol Oscillator, it is modified as:

$$\begin{cases} \ddot{K} + \left(-\tilde{M}_V + P_V\right)\dot{K} + \omega_V^2 K = 0 \\ M_V = a_V(1 - b_V K^2) + P_V \end{cases};$$
(A15)

b. For the Rayleigh Oscillator, it is modified as:

$$\begin{cases} \ddot{K} + \left(-\tilde{M}_R + P_R\right)\dot{K} + \omega_R^2 K = 0 \\ M_R = a_R\left(1 - b_R \dot{K}^2\right) + P_R \end{cases};$$
(A16)

c. For the Stuart-Landau Oscillator, it is modified as:

$$\begin{cases} \dot{Z} + \left(-\tilde{M}_S + P_S\right)Z = 0 \\ M_S = \sigma - \frac{l}{2}|Z|^2 + P_S \end{cases}.$$
(A17)

For each individual model listed above, $\tilde{M}_i$ (subscript $i$ represents V, R, and S, respectively) is the generalized muscle moment which is to be multiplied by the bell-shaped factor $(1 - Q)$ upon perturbation, and $P_i$ is the additional damping coefficient. Note that the minus sign prior to $M_i$ in the first equation of each set indicates that $M_i$ is a negative damping term that provides power to the

system, while $P_i$ is set positive for modeling the effect of bending toward the straight posture due to internal and external viscosity. Also note that *Equations A15-A17* would be equivalent to their original form (*Equations A11–A13*) when inhibition is absent (in this case, $\tilde{M}_i = M_i$).

By modeling the muscle inhibition process during locomotion, we were able to perform simulations of phase response experiments on individual models to produce perturbed systems dynamics (*Figure 5—figure supplement 1J–L*) and the corresponding PRCs (*Figure 5—figure supplement 1N–P* and *Figure 6—figure supplement 2*).

## Optimization of models

For each individual model we developed, the parameters were determined via a two-round fitting process. First, a subset of parameters was determined by fitting the model to observations of free-moving dynamics; the model could generate free-moving dynamics close to observations at this point. Second, the rest of the parameters were settled by fitting it to experimental phase response curves; a model would be fully determined at this point. Detailed descriptions of the two-step optimization procedure for individual models are provided as follows:

For the original threshold-switch model, parameters $\tau_u$, $M_0$, and $\tau_m$ were explicitly estimated from the experiments of free locomotion using phase portrait techniques described above. To simulate free locomotion, we further determined the position of switch points in the model (as indicated in *Figure 4C* red circle), which we did using method described by *Equation A7*. Next, we plugged the determined parameters into the model and conducted the second round of optimization by fitting the model with undetermined parameters $P_{th}$, $b$, as well as the parameters for simulating muscle inhibition — $H$ and $q$. We generated model PRC by perturbing the model oscillator at different times within a cycle period and settled the parameters such that the model PRC matched the experimental one with a minimum mean squared error (MSE) (During the computation of MSE, values of both model and experimental PRCs were sampled across the entire range of $\phi$ with 100 evenly distributed samples. In this case, $\Delta\phi = 2\pi/100$):

$$(P_{th}, b, H, q) = \underset{P_{th}, b, H, q}{\operatorname{argmin}} \sum_0^{2\pi} \left( PRC_{model}(P_{th}, b, H, q; \phi) - PRC_{experiment}(\phi) \right)^2 \Delta\phi \tag{A18}$$

To find the parameters that minimize the difference, a global minimum search was performed using the MATLAB function 'GlobalSearch' (*Ugray et al., 2007*). When run, the function repeatedly uses a local minimum solver with different batches of parameter range and attempts to locate a solution that produces the lowest MSE value.

Similarly, the two-step optimization procedures for individual alternative models are summarized in *Appendix 1—table 1*.

**Appendix 1—table 1.** Objective functions used in the optimization procedures for alternative models.

| Type | Free locomotion model | Complete model |
|---|---|---|
| van der Pol | $\underset{a_V, b_V, \omega_V}{\operatorname{argmin}} \left( \left( \frac{T_{vdP}}{T_{expt}} - 1 \right)^2 + \left( \frac{A_{vdP}}{A_{exp}} - 1 \right)^2 \right)$ | $\underset{p_V, H, q}{\operatorname{argmin}} \sum_0^{2\pi} \left( PRC_{vdp}(\phi) - PRC_{exp}(\phi) \right)^2 \Delta\phi$ |
| Rayleigh | $\underset{a_R, b_R, \omega_R}{\operatorname{argmin}} \left( \left( \frac{T_{Rayleigh}}{T_{expt}} - 1 \right)^2 + \left( \frac{A_{Rayleigh}}{A_{exp}} - 1 \right)^2 \right)$ | $\underset{p_R, H, q}{\operatorname{argmin}} \sum_0^{2\pi} \left( PRC_{Rayleigh}(\phi) - PRC_{exp}(\phi) \right)^2 \Delta\phi$ |
| Stuart-Landau | $\underset{a_S, b_S, \omega_S}{\operatorname{argmin}} \left( \left( \frac{T_{SL}}{T_{expt}} - 1 \right)^2 + \left( \frac{A_{SL}}{A_{exp}} - 1 \right)^2 \right)$ | $\underset{p_S, H, q}{\operatorname{argmin}} \sum_0^{2\pi} \left( PRC_{SL}(\phi) - PRC_{exp}(\phi) \right)^2 \Delta\phi$ |

Two-step optimization procedure for van der Pol, Rayleigh, and Stuart-Landau oscillators. The first-step optimization determines part of parameters such that individual models generate free locomotion dynamics. The second-step optimization leads to complete models such that models' perturbed dynamics and phase response curves are produced.

