## [Decision Letter]

**Acceptance summary:**

This paper combines behaviour quantification, optogenetic perturbation, and modelling to study *C. elegans* locomotion. It will be of interest to neuroscientists studying lcomotion and gait adaptation. The quantitative agreement between the model and experiments-importantly including the perturbation experiments-suggests forward locomotion in worms can be understood as being driven by a relaxation oscillator. This conclusion provides intuition for how worms move and should provide useful constraints on more detailed models that include neural anatomy and physiology.

**Decision letter after peer review:**

Thank you for submitting your article "Phase response analyses support a relaxation oscillator model of locomotor rhythm generation in *Caenorhabditis elegans*" for consideration by *eLife*. Your article has been reviewed by 3 peer reviewers, and the evaluation has been overseen by Manuel Zimmer as a Reviewing Editor and Ronald Calabrese as the Senior Editor. The following individual involved in review of your submission has agreed to reveal their identity: Carter Johnson (Reviewer #3).

The reviewers have discussed their reviews with one another, and the Reviewing Editor has drafted this to help you prepare a revised submission. As you can see below, the reviewers provide an extensive list of detailed comments, however the overall result of the reviews and subsequent discussions is that we are very excited about your work, and we would like to support publication in *eLife* of a revised manuscripts that addresses all comments. The comments below are partially concerned with similar issues but to retain each reviewers' details, we provide you here with the full list. Please submit with your revision a point-by-point response to the comments.

Essential revisions:*Reviewer #1:*

(R1.1) The authors set out to better understand the dynamics of *C. elegans* locomotion characterizing the rhythmic pattern of the worm's head during locomotion and by generating the phase response curve in that region. The authors then illustrate how well the relaxation-oscillator phenomenological model, with only a handful of free parameters optimized to match the data, is consistent with the observations, including observations under different mechanical loads.

(R1.2) The authors provide a characterization of the curvature dynamics in the worm's head region (0.1-0.3 body coordinate). The reasoning for focusing on this area in particular is not discussed or justified in the manuscript. Also, no comment is made whether the characteristics observed for the limit cycle in that region is consistent across the rest of the regions. From the experimental description and data shown (i.e., Figure 2B), it would appear that the authors had access to a characterization of the limit cycle across any region of the body of the worm, from head to tail. To avoid the appearance of cherry-picking a region, authors are encouraged to include the full characterization of the limit cycle for all the data available to them. In addition to that, authors are encouraged to clarify explicitly whether the characteristics observed in one region (e.g., nonsinusoidal limit cycle) hold true across the rest of the worm's body. How sinusoidal or nonsinusoidal, non-ellipsoidal, asymmetric is the limit cycle as a function of the body region of the worm, from head to tail? Crucially, if the characterization observed in the selected region is not equally pronounced across the rest of body, it seems crucial to discuss it in the interpretation of the results. This is particularly important given that throughout the manuscript, the authors interpret the results as applicable to the rhythmic generation across the whole body, not just that limited region (see first sentence of the Discussion, for example; and it is also relevant given that they discuss equally proprioceptive feedback in the head as well as in the rest of the body). These points apply to the characterization of the limit cycle given that the data is there but not shown; this is not the case for the phase response analysis, because understanding it for the full body would require further experimentation. However, this should be a point of discussion: is the expectation that the phase response curve through transient ontogenetic inhibition will look similar throughout the body? Is the same sawtooth-shape with the sharp transitions in the same part of the phase expected? What is the implication of the expected results if they are consistent or not consistent with the ones shown for the region focused on here?

(R1.2) The model assumes proprioceptive feedback from the curvature of the same region that is bending. However, the undifferentiated processes that are used as inspiration for this assumption extend posteriorly in the case of SMD and anteriorly in the case of B-class motor neurons. So the assumption that the curvature in one region can affect the bending in that same region should be discussed.

(R1.3) The model also assumes that the stretch receptors sense both the curvature and the rate of curvature. However, the authors provide no evidence of other *C. elegans* neurons where both of those can be sensed simultaneously. It seems feasible that a neuron might be sensitive to either the curvature or the rate of curvature, but it seems less feasible that the neuron would be sensitive to a linear combination of both. This seems to be a central assumption of the model, with little or no justification.

(R1.4) I commend the authors for comparing their selected model with other similar ones. I would highly encourage that the authors include a comparison of the models into the main manuscript. In particular, given the similarities between the results, I would highly encourage the authors to place the results of their chosen model next to the alternatives (it can be a figure like S10 but including the preferred model). As it is, with the alternatives in S10 and the rest of the figures from the best model in the main text, the comparison is significantly harder.

(R1.5) Finally, the discussion of previous relevant computational models of *C. elegans* locomotion is rather limited. There are at least four particularly relevant contributions that the authors do not discuss in relation to their own contribution:

(a) Johnson, Lewis, and Guy (2020) Neuromechanical Mechanisms of Gait Adaptation in *C. elegans*: Relative Roles of Neural and Mechanical Coupling.

(b) Izquierdo and Beer (2018) From head to tail: a neuromechanical model of forward locomotion in *Caenorhabditis elegans*.

(c) Kunert, Proctor, Brunton, Kutz (2017) Spatiotemporal feedback and network structure drive and encode *Caenorhabditis elegans* locomotion

(d) Denham, Ranner, Cohen (2018) Signatures of proprioceptive control in *Caenorhabditis elegans* locomotion.

*Reviewer #2:*

(R2.1) The paper's conclusions are supported by the data and the experiments and model results are clearly presented. In some cases there could be more analysis of uncertainty. For example, how consistent is the phase portrait? This could be illustrated by including some individual traces and/or by estimating a confidence interval (using resampling if necessary). A different kind of uncertainty that should be explained is that arising from fitting the model. Was the optimisation sensitive to the choice of starting parameters? Are there several local minima that explain the data reasonably well or did the optimisation always converge to the same parameter values?

(R2.2) The authors have focused on forward locomotion bouts and have also eliminated cycles with a period that deviated more than 20% from the average across worms within a session. How many of the bouts were eliminated with this threshold? Are any of the conclusions of the paper sensitive to this threshold?

(R2.3) The authors have convincingly ruled out the van der Pol and Stuart-Landau oscillators but the Rayleigh oscillator fits the data quite well with three parameters. Some quantification of the relative fit quality of the threshold-switch model and the Rayleigh might help guide future work.

(R2.4) Finally, the discussion does a good job of putting the results of the phenomenological threshold-switch model in the context of previously published detailed models but it might be strengthened with more quantitative comparisons. For example, do the observed muscle switching time scales or the nature of the proprioceptive threshold rule out or usefully constrain any of the detailed models?

(R2.5) The schematic figures (1 and 4) would be improved by leaving out the abbreviations. At a minimum, include (BWM) after 'body wall muscles' in the caption of figure 1.

(R2.6) Would a different colour map make Figure S2 clearer? Jet makes some regions stand out more than is warranted. For example, the purple patch in the lower right corner and red patch in the upper left corder are visually very salient but are probably just normal fluctuations within the noise.

(R2.7) In Figure 3, grey is used to indicate the control data in B but is used for individual trials in subsequent panels. Use a different colour for controls in B? Also add the grey curves to the legend next to D rather than only describing them in the caption.

(R2.8) Line 299, state the gene name of the GABA receptor.

(R2.9) In the Appendix there are a few cases of missing articles (for example line 812, "worm's head" should be "the worm's head" and on line 886 "animal's" should be "the animal's").

(R2.10) Starting on line 817, D^C is used to denote the distance from the origin over the normal cycle. To avoid confusion with the scaling constant c, perhaps choose a different symbol for the scaling constant or to indicate the normal cycle?*Reviewer #3:*

(R3.1) The main suggestion I have is to change the framing of the section "The Analysis of Alternative Models Supports a Relaxation Oscillation Mechanism'. In the first paragraph, the authors state that the purpose of this section is to "[ask] whether other models could also explain our findings." However, the section title and final paragraph suggest that the purpose of the section is to support the idea that the underlying oscillatory mechanism is a relaxation-oscillation. If my understanding of the authors' purpose here is correct, then I don't think the analysis of these three specific phenomenological models provides evidence of this. Specifically, the last paragraph of this section (Lines 586-590), suggests that because the non-relaxation oscillator (Stuart-Landau) was not able to capture your results (non-sinusoidal limit cycle and sawtooth PRCs), the underlying mechanism is a relaxation-oscillation. I don't think these models support that claim, because a different non-relaxation oscillator tuned precisely may be able to capture these results. I think a much more thorough analysis of general oscillator dynamics would be needed to make such a claim. To be clear, I think the paper is strong enough without this analysis, I would just be more cautious about the claims here.

(R3.2) The strength of the author's original computational model is that it is based on specific mechanisms thought to underlie *C. elegans* neurolocomotion (specifically motor neurons, muscles, and proprioception). The authors discuss how this model functions as a relaxation-oscillator in the discussion, but I think it would be stronger in this section.

(R3.3) Intro or discussion – In addition to Cohen's group's models, two recent modeling papers from other modeling groups have investigated the *C. elegans* neurolocomotion system explicitly as a chain of coupled neuromechanical oscillators (Olivares et al., 2021 doi:10.3389/fncom.2021.572339 and Johnson et al., 2021 doi:10.1137/20M1346122). In particular, Johnson et al., 2021 show how phase-response properties influence coordination of the neuromechanical oscillators. Describing how your computational model compares with or supports these other recent models would help give context to the modeling contribution of this work.

(R3.14) Lines 311-313: The authors mention that sawtooth PRCS "may reflect a phase-resetting property of an oscillator with respect to a perturbation". I think more detail would be helpful here. To my knowledge, this is referring to the idea that if an oscillator gets a really strong perturbation, the phase-response curve turns into a "phase-resetting curve", where the large perturbation essentially resets the phase of the oscillator, and then it just marches forward linearly through time. Are you suggesting that this sawtooth shape is indicative of your perturbation being large? This would be an important detail to include.

(R3.5) Section "Analysis of Alternative Models Supports a Relaxation Oscillation Mechanism":

The strength of the author's original computational model is that it is based on specific mechanisms thought to underlie *C. elegans* neurolocomotion (specifically motor neurons, muscles, and proprioception). To support the idea that the underlying mechanism is a relaxation-oscillation, I would suggest that the authors instead analyze their own computational model and show or point out how it functions structurally as a relaxation-oscillation. This is mentioned briefly in the discussion, but I think it would be stronger in this section. Furthermore, I think the authors could give the reader a clearer reason why exactly showing that the mechanism is a relaxation-oscillation is important. To my understanding, the open question in the *C. elegans* locomotion-modeling literature is whether the oscillations are fundamentally neurally-driven (like a CPG or HCO) or reflex-driven (like the proprioceptive mechanism here). Perhaps evidence for the relaxation-oscillation mechanism could be made by considering an alternative mechanistic model with neurally-driven oscillations and whether it can capture the new phase response data.

(R3.6) The authors have not given adequate description of the model fitting process in the main text. In line 548, you only mention that appropriate parameters were chosen, and reference the supplement. I don't think too much detail is needed here, but explaining that the models were fit to match both the limit cycles and PRCs would be helpful here. My first impression of this section was that the limit cycles were matched and the PRCs were emergent. After looking at the supplementary details I saw that you explicitly attempted to match both results. This would be helpful to put up front.

(R3.7) Lines 61-64: Sentence grammar/structure. The "but also …" is missing a "not only" earlier in the sentence (e.g. "A comprehensive understanding of animal locomotion should therefore encompass not only neural activity, muscle activity, and sensory feedback, but also biomechanical forces within the animal's body and between the animal and its environment (Figure 1A; 1-3).").

(R3.8) Lines 152-153 and Figure 2D caption: I think it would help to clarify in either or both of these places that the boxed portion of the nematode bodies corresponds to the curvature region (0.1-0.3 body coordinates).

(R3.9) Lines 940-951: The mechanics here depend on the assumption of a traveling wave of fixed wavelength λ down the body. I would mention explicitly that both the wavelength λ and normal drag coefficient CN will be varied with fluid viscosity.

---

## [Author Response]

Essential revisions:Reviewer #1:(R1.1) The authors set out to better understand the dynamics of *C. elegans* locomotion characterizing the rhythmic pattern of the worm's head during locomotion and by generating the phase response curve in that region. The authors then illustrate how well the relaxation-oscillator phenomenological model, with only a handful of free parameters optimized to match the data, is consistent with the observations, including observations under different mechanical loads.(R1.2) The authors provide a characterization of the curvature dynamics in the worm's head region (0.1-0.3 body coordinate). The reasoning for focusing on this area in particular is not discussed or justified in the manuscript. Also, no comment is made whether the characteristics observed for the limit cycle in that region is consistent across the rest of the regions. From the experimental description and data shown (i.e., Figure 2B), it would appear that the authors had access to a characterization of the limit cycle across any region of the body of the worm, from head to tail. To avoid the appearance of cherry-picking a region, authors are encouraged to include the full characterization of the limit cycle for all the data available to them. In addition to that, authors are encouraged to clarify explicitly whether the characteristics observed in one region (e.g., nonsinusoidal limit cycle) hold true across the rest of the worm's body. How sinusoidal or nonsinusoidal, non-ellipsoidal, asymmetric is the limit cycle as a function of the body region of the worm, from head to tail? Crucially, if the characterization observed in the selected region is not equally pronounced across the rest of body, it seems crucial to discuss it in the interpretation of the results. This is particularly important given that throughout the manuscript, the authors interpret the results as applicable to the rhythmic generation across the whole body, not just that limited region (see first sentence of the Discussion, for example; and it is also relevant given that they discuss equally proprioceptive feedback in the head as well as in the rest of the body).

As suggested, we have calculated the limit cycles of bending dynamics as a function of body coordinate from the head to the tail and included the results in the manuscript. We also provide a justification for focusing on the head region for most of our analyses (see Lines 159-162 and Figure 2—figure supplement 1).

These points apply to the characterization of the limit cycle given that the data is there but not shown; this is not the case for the phase response analysis, because understanding it for the full body would require further experimentation. However, this should be a point of discussion: is the expectation that the phase response curve through transient ontogenetic inhibition will look similar throughout the body? Is the same sawtooth-shape with the sharp transitions in the same part of the phase expected? What is the implication of the expected results if they are consistent or not consistent with the ones shown for the region focused on here?

We conducted further experiments to optogenetically inhibit body wall muscles in other regions of a worm besides the head region. The corresponding PRCs are now shown and discussed in the manuscript (see Lines 265-269, and Lines 952-978 and Figure 3—figure supplement 5 in Appendix).

(R1.2) The model assumes proprioceptive feedback from the curvature of the same region that is bending. However, the undifferentiated processes that are used as inspiration for this assumption extend posteriorly in the case of SMD and anteriorly in the case of B-class motor neurons. So the assumption that the curvature in one region can affect the bending in that same region should be discussed.

The threshold-switch model as well as the other models are developed based on experimental observations of the head region, defined as the body coordinate interval between 0.1 and 0.3, which is about 200 μm long. The asynaptic processes in the posterior portion of the SMD neurons are ~40 μm long (c.f. the graphical abstract for Reid et al., Neuroscience 2015, PMID: 26480814) and have been covered by the region of our analyses. The asynaptic processes in the anterior portion of the near-head Btype motor neurons are 100~200 μm long (see Figure 1 in Chen et al., PNAS 2006, PMID: 16537428) where most of the processes have also been covered by the region of our analyses. Therefore, the processes of the candidate proprioceptors (SMD and anterior B-type motor neurons) could affect the bending in the analyzed region.

(R1.3) The model also assumes that the stretch receptors sense both the curvature and the rate of curvature. However, the authors provide no evidence of other *C. elegans* neurons where both of those can be sensed simultaneously. It seems feasible that a neuron might be sensitive to either the curvature or the rate of curvature, but it seems less feasible that the neuron would be sensitive to a linear combination of both. This seems to be a central assumption of the model, with little or no justification.

First, it does not seem at all implausible to us that a *C. elegans* neuron might be sensitive to a combination of curvature and rate of change of curvature. For example, in rat dorsal root ganglia, rapidly-adapting (RA) and intermediate-adapting (IA) mechanosensors encode a combination of stimulus size and velocity. See for example Figure 1 in Rugiero et al., J Physiol 2010 (PMID: 19948656).

Second, even if we assume a single neuron is not sensitive to a combination of curvature and rate of change of curvature, a collection of neurons involved in motor rhythm generation may be. The stretch receptors in our model might be considered as a collective behavior of curvature-sensing neurons and curvature-change-sensing

neurons.

We have revised this discussion to clarify these points (see Lines 714-729).

(R1.4) I commend the authors for comparing their selected model with other similar ones. I would highly encourage that the authors include a comparison of the models into the main manuscript. In particular, given the similarities between the results, I would highly encourage the authors to place the results of their chosen model next to the alternatives (it can be a figure like S10 but including the preferred model). As it is, with the alternatives in S10 and the rest of the figures from the best model in the main text, the comparison is significantly harder.

As suggested, we have added our primary model to the comparison between all models (see Figure 5—figure supplement 1 and Figure 6—figure supplement 2).

(R1.5) Finally, the discussion of previous relevant computational models of *C. elegans* locomotion is rather limited. There are at least four particularly relevant contributions that the authors do not discuss in relation to their own contribution:(a) Johnson, Lewis, and Guy (2020) Neuromechanical Mechanisms of Gait Adaptation in *C. elegans*: Relative Roles of Neural and Mechanical Coupling.(b) Izquierdo and Beer (2018) From head to tail: a neuromechanical model of forward locomotion in *Caenorhabditis elegans*.(c) Kunert, Proctor, Brunton, Kutz (2017) Spatiotemporal feedback and network structure drive and encode *Caenorhabditis elegans* locomotion(d) Denham, Ranner, Cohen (2018) Signatures of proprioceptive control in *Caenorhabditis elegans* locomotion.

We thank the reviewer for these suggestions. These models have been cited and discussed in the revised manuscript (see Lines 117-119, 639-640, 646-648, and 672-674).

Reviewer #2:(R2.1) The paper's conclusions are supported by the data and the experiments and model results are clearly presented. In some cases there could be more analysis of uncertainty. For example, how consistent is the phase portrait? This could be illustrated by including some individual traces and/or by estimating a confidence interval (using resampling if necessary).

As suggested, we have added 10 randomly selected individual traces to the average

trace shown in Figure 2D.

A different kind of uncertainty that should be explained is that arising from fitting the model. Was the optimisation sensitive to the choice of starting parameters? Are there several local minima that explain the data reasonably well or did the optimisation always converge to the same parameter values?

Our fitting procedure for optimizing model parameters was not sensitive to the initial choices. For each individual model developed in this manuscript, the parameters were determined via a two-round fitting procedure as discussed in Results (Lines 367-373) and Appendix (Lines 1217-1248). In the first round, a model’s parameters were optimized to match the experimental free locomotion, using cost functions shown in Equation S7 (for threshold-switch model) and Table S1 (for alternative models). During this round, the parameters converged to a unique set of values since there is only one local minimum within the range of parameters (a large range of parameters that could generate oscillation). In the second round, the remaining undetermined parameters were optimized to match the experimental PRC curve using a global minimum search strategy within a range of parameters. Since it is a global search algorithm, the optimization is independent of the choice of initial parameters.

(R2.2) The authors have focused on forward locomotion bouts and have also eliminated cycles with a period that deviated more than 20% from the average across worms within a session. How many of the bouts were eliminated with this threshold?

Under the threshold 20%, 857 out of 1910 trials (~45%) that deviated from the average period were eliminated.

Are any of the conclusions of the paper sensitive to this threshold?

None of the conclusions of the paper are sensitive to the threshold. We measured the PRC and limit cycle under thresholds up to 50% (343 out of 1910 bouts (~18%) were dropped), and down to 10% (1270 out of 1910 bouts (67%) were dropped). We find that the resulting curves are largely independent of the choice of threshold (see Author response image 1).

(R2.3) The authors have convincingly ruled out the van der Pol and Stuart-Landau oscillators but the Rayleigh oscillator fits the data quite well with three parameters. Some quantification of the relative fit quality of the threshold-switch model and the Rayleigh might help guide future work.

As now shown in Figure 5—figure supplement 1, the threshold-switch model fits the experimental PRC better than Rayleigh model by a considerable margin in terms of mean square error (threshold-switch: 0.12; vdP: 0.21; Rayleigh: 0.37).

(R2.4) Finally, the discussion does a good job of putting the results of the phenomenological threshold-switch model in the context of previously published detailed models but it might be strengthened with more quantitative comparisons. For example, do the observed muscle switching time scales or the nature of the proprioceptive threshold rule out or usefully constrain any of the detailed models?

As suggested, we have added an extensive discussion to acknowledge the potential usefulness of parameters measurements for constraining detailed models (see Lines 660-674).

(R2.5) The schematic figures (1 and 4) would be improved by leaving out the abbreviations. At a minimum, include (BWM) after 'body wall muscles' in the caption of figure 1.

We have added the full names for all the abbreviations shown in the Figures (1 and 4) and the corresponding captions.

(R2.6) Would a different colour map make Figure S2 clearer? Jet makes some regions stand out more than is warranted. For example, the purple patch in the lower right corner and red patch in the upper left corder are visually very salient but are probably just normal fluctuations within the noise.

We have changed this figure to have a cyclic color map with a constant lightness which avoids visual emphasis on certain parts of the fluctuations (see Figure 3— figure supplement 2).

(R2.7) In Figure 3, grey is used to indicate the control data in B but is used for individual trials in subsequent panels. Use a different colour for controls in B? Also add the grey curves to the legend next to D rather than only describing them in the caption.

As suggested, we have revised Figure 3 and its caption.

(R2.8) Line 299, state the gene name of the GABA receptor.

We have added the gene information in the main text (Lines 307-309) and the figure captions (Figure 3—figure supplement 8). Also, strain names that previously appeared in figure captions have been moved to Methods.

(R2.9) In the Appendix there are a few cases of missing articles (for example line 812, "worm's head" should be "the worm's head" and on line 886 "animal's" should be "the animal's")

We have corrected the multiple missing articles pointed out by the reviewer.

(R2.10) Starting on line 817, D^C is used to denote the distance from the origin over the normal cycle. To avoid confusion with the scaling constant c, perhaps choose a different symbol for the scaling constant or to indicate the normal cycle?

We have used a different symbol for the scaling constant (see Figure 3—figure supplement 2 and Lines 838-867).

Reviewer #3:(R3.1) The main suggestion I have is to change the framing of the section "The Analysis of Alternative Models Supports a Relaxation Oscillation Mechanism'. In the first paragraph, the authors state that the purpose of this section is to "[ask] whether other models could also explain our findings." However, the section title and final paragraph suggest that the purpose of the section is to support the idea that the underlying oscillatory mechanism is a relaxation-oscillation. If my understanding of the authors' purpose here is correct, then I don't think the analysis of these three specific phenomenological models provides evidence of this. Specifically, the last paragraph of this section (Lines 586-590), suggests that because the non-relaxation oscillator (Stuart-Landau) was not able to capture your results (non-sinusoidal limit cycle and sawtooth PRCs), the underlying mechanism is a relaxation-oscillation. I don't think these models support that claim, because a different non-relaxation oscillator tuned precisely may be able to capture these results. I think a much more thorough analysis of general oscillator dynamics would be needed to make such a claim. To be clear, I think the paper is strong enough without this analysis, I would just be more cautious about the claims here.

We agree that our investigation of a few alternative models may not lead to the conclusion claimed by the original section title. We have rephrased this section and changed its title appropriately (see Lines 551-608).

(R3.2) The strength of the author's original computational model is that it is based on specific mechanisms thought to underlie *C. elegans* neurolocomotion (specifically motor neurons, muscles, and proprioception). The authors discuss how this model functions as a relaxation-oscillator in the discussion, but I think it would be stronger in this section.

See response to R3.5.

(R3.3) Intro or discussion – In addition to Cohen's group's models, two recent modeling papers from other modeling groups have investigated the *C. elegans* neurolocomotion system explicitly as a chain of coupled neuromechanical oscillators (Olivares et al., 2021 doi:10.3389/fncom.2021.572339 and Johnson et al., 2021 doi:10.1137/20M1346122). In particular, Johnson et al., 2021 show how phase-response properties influence coordination of the neuromechanical oscillators. Describing how your computational model compares with or supports these other recent models would help give context to the modeling contribution of this work.

As suggested, we have discussed and cited these two papers in the Introduction (Lines 117-119) and Discussion (Lines 639-641, 646-648, and 672-674).

(R3.14) Lines 311-313: The authors mention that sawtooth PRCS "may reflect a phase-resetting property of an oscillator with respect to a perturbation". I think more detail would be helpful here. To my knowledge, this is referring to the idea that if an oscillator gets a really strong perturbation, the phase-response curve turns into a "phase-resetting curve", where the large perturbation essentially resets the phase of the oscillator, and then it just marches forward linearly through time. Are you suggesting that this sawtooth shape is indicative of your perturbation being large? This would be an important detail to include.

We did not intend the term phase resetting to be different from phase response in our manuscript. We apologize for any confusion and have revised this passage.

To the main point of the question, we did also perform the PRC experiments with weaker stimuli (0.055 s muscle inhibition, Figure 3—figure supplement 4), which resulted in a similar sawtooth-shaped PRC.

(R3.5) Section "Analysis of Alternative Models Supports a Relaxation Oscillation Mechanism":The strength of the author's original computational model is that it is based on specific mechanisms thought to underlie *C. elegans* neurolocomotion (specifically motor neurons, muscles, and proprioception). To support the idea that the underlying mechanism is a relaxation-oscillation, I would suggest that the authors instead analyze their own computational model and show or point out how it functions structurally as a relaxation-oscillation. This is mentioned briefly in the discussion, but I think it would be stronger in this section.

In Discussion (Lines 675-680 and 694-702), we have explained how our model satisfies the properties of a relaxation oscillator. In the caption of Figure 4A and Appendix (Lines 1018-1030), each site respectively contains a step-by-step explanation of how our model is constructed as a relaxation oscillator.

Furthermore, I think the authors could give the reader a clearer reason why exactly showing that the mechanism is a relaxation-oscillation is important. To my understanding, the open question in the *C. elegans* locomotion-modeling literature is whether the oscillations are fundamentally neurally-driven (like a CPG or HCO) or reflex-driven (like the proprioceptive mechanism here). Perhaps evidence for the relaxation-oscillation mechanism could be made by considering an alternative mechanistic model with neurally-driven oscillations and whether it can capture the new phase response data.

As highlighted in Introduction/Discussion (Lines 120-125, 655-659, and 725-729) and described in Results/Appendix (Figure 4A, Lines 349-373 and 985-1041), our model is designed to be a phenomenological model that does not specify the neural and synaptic details of the motor circuit but rather capture the essential mechanisms underlying motor rhythm generation. We have compared our model with several alternatives based on other types of oscillators. We feel that consideration of alternative models with neurally-driven oscillations would be interesting but beyond the scope of the work.

In our view, the importance of showing that *C. elegans* locomotion operates by a relaxation-oscillation mechanism is reflected by demonstrating how and why such a simple model with a few parameters could capture several different experimental observations (free-locomotion, phase responses, and gait adaptation to external load), and in the potential to obtain an integrative understanding of how an organism

generates behavior.

(R3.6) The authors have not given adequate description of the model fitting process in the main text. In line 548, you only mention that appropriate parameters were chosen, and reference the supplement. I don't think too much detail is needed here, but explaining that the models were fit to match both the limit cycles and PRCs would be helpful here. My first impression of this section was that the limit cycles were matched and the PRCs were emergent. After looking at the supplementary details I saw that you explicitly attempted to match both results. This would be helpful to put up front.

As suggested, we have moved the two-round fitting procedure to the beginning of the corresponding Result section to better clarify this process (see Lines 369-373). More details can also be found in the Appendix (Lines 1217-1248)

(R3.7) Lines 61-64: Sentence grammar/structure. The "but also …" is missing a "not only" earlier in the sentence (e.g. "A comprehensive understanding of animal locomotion should therefore encompass not only neural activity, muscle activity, and sensory feedback, but also biomechanical forces within the animal's body and between the animal and its environment (Figure 1A; 1-3).").

Thanks for pointing out this grammatical error. We have corrected this.

(R3.8) Lines 152-153 and Figure 2D caption: I think it would help to clarify in either or both of these places that the boxed portion of the nematode bodies corresponds to the curvature region (0.1-0.3 body coordinates).

As suggested, we have clarified the contents in both places that the boxed region indicates the anterior region of the worm body (0.1-0.3 body coordinates).

(R3.9) Lines 940-951: The mechanics here depend on the assumption of a traveling wave of fixed wavelength λ down the body. I would mention explicitly that both the wavelength λ and normal drag coefficient CN will be varied with fluid viscosity.

As suggested, we have modified this subsection to acknowledge that the derivation of the mechanics depends on the assumption of a travelling wave with a fixed wavelength (see Lines 1004-1017). The dependence of the wavelength and drag coefficient on the viscosity has been noted after the derivation.